# BELT-2: Bootstrapping EEG-to-Language representation alignment for multi-task brain decoding

## Abstract

The remarkable success of large language models (LLMs) across various multi-modality applications is well established. However, integrating large language models with humans, or brain dynamics, remains relatively unexplored. In this paper, we introduce BELT-2, a pioneering multi-task model designed to enhance both encoding and decoding performance from EEG signals. To bolster the quality of the EEG encoder, BELT-2 is the first work to innovatively 1) adopt byte-pair encoding (BPE)-level EEG-language alignment and 2) integrate multi-task training and decoding in the EEG domain. Inspired by the idea of ***Bridging the Brain with GPT***, we further connect the multi-task EEG encoder with LLMs by utilizing prefix-tuning on intermediary output from the EEG encoder. These innovative efforts make BELT-2 a pioneering breakthrough, making it the first work in the field capable of decoding coherent and readable sentences from non-invasive brain signals. Our experiments highlight significant advancements over prior techniques in both quantitative and qualitative measures, achieving a decoding performance with a BLEU-1 score of 52.2% on the ZuCo dataset. Furthermore, BELT-2 shows a remarkable improvement ranging from 31% to 162% on other translation benchmarks. Codes can be accessed via the provided anonymous link [1].

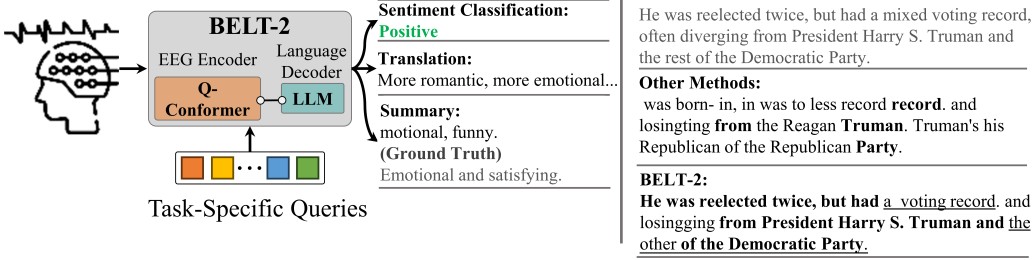

Figure 1: Overview of BELT-2. The first work of multi-task brain decoding by bridging the Q-Conformer EEG encoder and LLMs. Provided samples also suggest BELT-2 is the first to achieve fluent sentence decoding results from noninvasive brain signals.

# 1 Introduction

Recently, the emergence of large language models (LLMs) has spurred efforts to integrate them with various modalities, such as VisualLLMs (Liu et al., 2023b; Oquab et al., 2023), and Robotics (Driess et al., 2023). These methods achieved remarkable improvement in various task settings. Yet, an important topic, the direct combination of LLMs with human intention remains relatively unexplored. Nonetheless, the inherent subject-wise non-stationary characteristics of Electroencephalography (EEG) signals, coupled with rigorous experimental protocols, make the task of decoding words or sentences exceptionally challenging.

[1] https://anonymous.4open.science/r/BELT-2-0048

Explorations on brain-to-text and brain-to-speech decoding in the earlier stage (Herff et al., 2015; Makin et al., 2020; Panachakel & Ramakrishnan, 2021; Nieto et al., 2021) mostly perform decoding on a closed word-level set, which still has notable restrictions on vocabulary size and limitations to more intricate application scenarios. For the brain-to-language decoding, EEG-to-Text (Wang & Ji, 2022) introduced the open-vocabulary decoding of EEG signals with an initial performance baseline. DeWave (Duan et al., 2023) improved decoding performance by introducing a discrete encoder for EEG. BELT (Zhou et al., 2023a) which boosted decoding performance by leveraging language supervision. However, these methods are limited to single-task settings and have not achieved multi-task decoding from brain signals to natural languages. An extensive **related works** is provided in Appendix A due to space limit.

In this paper, we propose BELT-2, the first EEG-language learning framework to bridge the modality gap and effectively exploit LLM's generative capacity for EEG decoding. BELT-2 enhances three key aspects of brain decoding research. 1) It is the first to introduce **BPE-level contrastive learning** for EEG-to-language alignment. 2) It first introduces a **prompt-based multi-task encoder** for EEG research. 3) It proposes a cost-effective solution for connecting an EEG encoder with a large language model (LLM).

More specifically, we introduce a novel discrete querying conformer (Q-Conformer) as the EEG encoder to improve encoding capacity and enable multitasking (Figure 5). Unlike previous single-task EEG encoders (Zhou et al., 2023a; Duan et al., 2023), Q-Conformer is able to extract task-specific contexts according to a given query prompt. For the training of Q-Conformer, we propose the BPE-level EEG-language contrastive learning (BPE-CL) to bootstrap the learning of language-aligned EEG representation. After training, we bridge the Q-Conformer and an LLM decoder by prefix-tuning with both models frozen. To improve the performance of the briding, we further propose a technique called speculative augmentation (SA) to improve the training efficiency. The main contributions of BELT-2 could be concluded in four aspects.

- This paper presents a novel framework capable of decoding fluent open-vocabulary sentences, facilitating multi-task EEG decoding including EEG translation, sentiment classification, and summarization.
- The Q-Conformer is proposed to improve the encoding ability and the scalability for multi-tasking while the BPE-level contrastive learning establishes a firm alignment between EEG and language representations.
- This paper provides a cost-effective bridging method for connecting LLMs with brain encodings by turning virtual-prefix. A speculative augmentation method is introduced to further improve the bridging performance.
- Experimental results suggest that the proposed BELT-2 exceeds SOTA performance on different EEG decoding tasks. For EEG translation, BELT-2 achieves 52.59 BLEU-1, 17.85 BLEU-4, and 40.1 Rouge-1 Precision, which significantly outperforms the previous baseline by 31%, 162% and 26% respectively. On sentiment classification, BELT-2 achieves 74.62% accuracy without further assistance from additional classifiers or external datasets. BELT-2 is also the first work that achieves EEG summarization with a SOTA 31.17 BLEU-1 score.

## 2 BELT-2

BELT-2 introduces the Q-Conformer which enhances both the capacity to encode EEG information and the extendibility to multi-task. To bridge the modality gap between EEG and language, we boost EEG-to-Language representation learning through two learning stages: (1) the EEG-to-language alignment learning stage for learning the Q-Conformer EEG encoder. (2) a prefix-tuning stage for bridging Q-Conformer with LLM.

### 2.1 Q-CONFORMER AS EEG ENCODER

The overall structure of the Q-Conformer is illustrated in Figure 5 which consists of a discrete conformer, a Context Transformer (C-Former), and a query prompt. The discrete conformer functions as a discrete EEG tokenizer that captures primitive patterns from the input EEG embeddings. The C-Former extracts mid-layer coding (MLC) that contains context information specific to a given task given by the learnable query prompt.

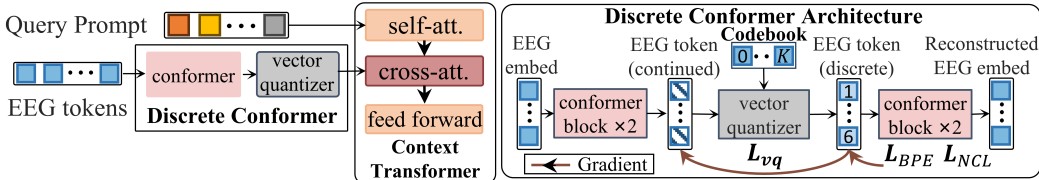

Figure 2: The overall structure of the Q-Conformer. It consists of a discrete conformer, a context transformer (C-Former), and a query prompt. The input EEG embeddings (EEG embed) are first processed by the conformer into continuous EEG tokens. A vector quantizer is then used to discretize the EEG tokens. Then, a query prompt interacts with the discrete EEG token via the cross-attention layer from in the C-Former to extract task-specific context information from the discrete EEG tokens.

**Discrete Conformer:** The discrete conformer consists of a conformer model and a vector quantizer. After preprocessing, the raw EEG waveform is segmented into windows using eye-tracking information. Then a frequency domain transform converts EEG segments into fix-size EEG embeddings $\mathbf{e} \in \mathbb{R}^{L \times N \times D}$. $L$ is the maximum length of the embedding sequence, $N$ denotes the number of EEG channels, and $D$ denotes the embedding size. The conformer model consists of 2 conformer blocks which follow the structure manifested in (Gulati et al., 2020). The conformer model $E(\cdot)$ converts the EEG embeddings $\mathbf{e}$ into continuous EEG tokens $\mathbf{h} \in \mathbb{R}^{L \times N \times d}$, where $d$ denotes the size of the continuous EEG tokens.

We then convert $\mathbf{h}$ to a set of discrete tokens $\mathbf{b}$ by a vector quantizer (VQ) that looks up the nearest discrete code $\mathbf{v}_k, k = \{0, 1, \cdots, K\}$ from the codebook $\mathcal{V}$ (Razavi et al., 2019). The quantization process $\mathbf{z}_q(\mathbf{h})$ can be written as Equation 1.

$$\mathbf{z}_q(\mathbf{h}) = \{\mathbf{z}_q(\mathbf{h_i})\}_{i=0}^{L}, \quad \mathbf{z}_q(\mathbf{h_i}) = \mathbf{v}_k, \quad k = \arg\min_{j} \|\mathbf{h}_j - \mathbf{v}_j\|_2^2 \quad (1)$$

We use $L_{vq}$ (Equation 2) to train the discrete codebook. The $L_{vq}$ is a weighted summation of 4 loss terms. The first two terms are the codebook loss and the commitment loss. They are used to update the codebook by minimizing the information loss between the input and the output discrete tokens Van Den Oord et al. (2017). The third term encourages the balanced use of all entries in the codebook and prevents codebook collapse during training (Dieleman et al., 2018). The last term is a reconstructive loss that ensures the information passed to the VQ is sufficient to describe the EEG signal.

$$\mathcal{L}_{vq} = \|sg\,[\mathbf{h}] - \mathbf{z}_q(\mathbf{h})\|_2^2 + \|\mathbf{h} - sg\,[\mathbf{z}_q(\mathbf{h})]\|_2^2 + \frac{1}{|\mathcal{V}|} \sum_{k=0}^{|\mathcal{V}|} p_k \log p_k + \|\mathbf{e} - \hat{\mathbf{e}}\|_2^2 \quad (2)$$

, where $sg\,[\cdot]$ stands for the stop-gradient operator which is an identity at the forward pass while having zero gradients during the backward pass. $|\mathcal{V}|$ denotes the size of the discrete codebook and $p_k$ denotes the softmax probability of the codebook entry $k$ being used in each batch. $\hat{\mathbf{e}}$ denotes the reconstructed EEG embedding from $\mathbf{z}_q(\mathbf{h})$ using 2 comformer blocks.

**C-Former and Query Prompt** We create a set number of learnable query embeddings (query prompt) as input to the C-Former. The C-Former is composed of self-attention layers and cross-attention layers arranged in consecutive order. After feeding the query prompts and the discrete EEG tokens into the C-Former, the query prompts interact with each other through the self-attention layers and further interact with the discrete EEG tokens through the following cross-attention layer. A new query prompt will be initialized when training the Q-Conformer for a specific task. After training on a specific task, the query prompts learn to act as the instruction of the current task that guides the C-Former to extract MLC as the task-specific context from the EEG modality.

This querying mechanism enables a more flexible adaptation of the pretrained Q-Conformer to a new downstream task by adding a new set of query prompts. It also allows the reusing of knowledge learned from previous training tasks. In our experiment setup, we initialize the C-Former with the pre-trained weights of BART$_{large}$ (Lewis et al., 2019). We employ a query prompt of 20 learnable tokens for a specific, with each query possessing a dimensionality of 1024.

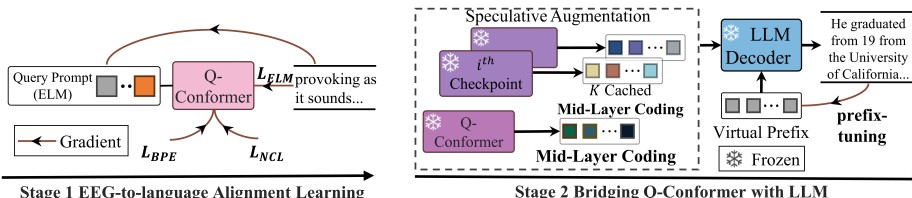

Figure 3: BELT-2's two-stage training schema. For EEG-to-language alignment learning (**left**), we jointly optimize three objectives that firmly establish the EEG-to-language alignment and enforce the query prompt to extract the EEG context most relevant to a task. For bridging of Q-Conformer and LLM (**right**), connect a frozen EEG model (Q-Conformer) and a frozen LLM by tuning the continuous virtual prefix using the prefix-tuning method. Speculative augmentation is used to boost the performance of the prefix-tuning process.

## 2.2 EEG-TO-LANGUAGE ALIGNMENT LEARNING

In the EEG-to-language alignment learning stage, we train the Q-Conformer and align the encoded EEG tokens to the language modality. To achieve EEG-to-Language alignment, we combine two contrastive objectives and a pretraining objective to the VQ objective in Equation 2. The two contrastive objectives include (1) BPE-level contrastive learning (BPE-CL), and (2) Negative Contrastive learning (NCL). We further pretrain the Q-Conformer to achieve a task-specific query prompt by the EEG-to-Language matching (ELM) objective, which guides the C-Former to extract MLC that contains the most relevant EEG contexts in the specific task.

**BPE-level contrastive learning** (BPE-CL) learns to align the discrete EEG tokens with BPE subword embeddings by maximizing their mutual information. Unlike the BELT-1 model (Zhou et al., 2023a) where contrastive learning is only performed at the word level, we perform EEG-language alignment in the BPE subword level to improve EEG-language alignment. Given the limited size of EEG-language pairs in the training set, this method enforces stronger semantic guidance to the EEG representation while enhancing the matching of subword units that are out-of-training vocabulary.

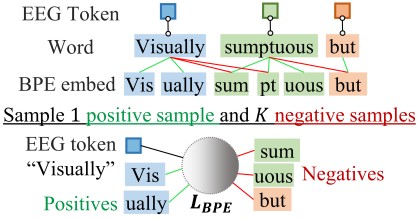

Figure 4: The illustration of BPE-level contrastive learning.

The sampling strategy of the BPE-CL is illustrated in Figure 4. We commence by converting words into BPE tokens $\mathbf{w} \in \mathcal{W}$, e.g., converting "$Visually$" into ["$Vis$", "$ually$"]. The embeddings of these BPE tokens serve as positive targets for the EEG token corresponding to "$Visually$" while BPE tokens other words are viewed as negative targets. We uniformly sample 1 positive target and $K$ negative targets for each discrete EEG token in a training batch. The learning objective $L_{bpe}$ for the discrete EEG tokens and the BPE embeddings is formulated as:

$$\mathcal{L}_{bpe} = -\log \frac{\exp\left(\mathbf{z}_q(\mathbf{h})^\top \mathbf{w}^+\right)}{\exp\left(\mathbf{z}_q(\mathbf{h})^\top \mathbf{w}^+\right) + \sum_{i=1}^{K} \exp\left(\mathbf{z}_q(\mathbf{h})^\top \mathbf{w}^-\right)}, \tag{3}$$

, where $\mathbf{w}^+$ is the sampled embedding of the positive BPE token and $\mathbf{w}^-$ is the negative ones.

**Negative contrastive learning** (NCL) aims to further improve the distinctions between the discrete EEG tokens by randomly sampling $K$ negative EEG tokens as distractors for each discrete EEG token in a training batch, which is defined as:

$$\mathcal{L}_{neg} = -\log \frac{1}{\sum_{i=1}^{K} \exp\left(\mathbf{z}_q(\mathbf{h})^\top \mathbf{z}_q(\mathbf{h})^-\right)}, \tag{4}$$

, where $\mathbf{z}_q(\mathbf{h})^-$ are sampled negative tokens from the batch and $\mathbf{z}_q(\mathbf{h})$ is defined in Equation 1. This objective enlarges the distinction among EEG tokens that are indistinguishable upon reading different words, easing the decoding effort.

**EEG-to-language matching** (ELM) aims to function as the pretraining task for learning the initial task-specific query prompt, which in terms is used to instruct the C-Former to extract task-specific context from the EEG tokens. We use a sequence-to-sequence machine translation loss similar to previous works Zhou et al. (2023a); Wang & Ji (2022); Duan et al. (2023) as the objective function. Given the word-level EEG embedding sequence and text sentence pair $\langle \mathcal{E}, \mathcal{S} \rangle$, we maximize the probability of the decoded sentence $p(\mathcal{S}|\mathcal{E})$ produced by the Q-Conformer. The learning objective is a machine translation term $L_{tr}$, which could be written as follows:

$$\mathcal{L}_{elm} = -\sum_{l}^{L} \log p(s_l \in \mathcal{S}|\mathbf{q}) \tag{5}$$

, where $L$ is the total length of the target text sequence, $s_l \in \mathcal{S}$ denotes the decoded tokens from the C-Former and $\mathbf{q}$ denotes the query prompt.

## 2.3 Bridging Q-Conformer with LLM

We propose to bridge the frozen Q-Conformer and a frozen LLM to leverage both models effectively for EEG-to-Language tasks by tuning a set of virtual prefixes added to the output embeddings of the Q-Conformer, in order to achieve stronger performance at a lower training cost.

**Prefix-tuning**  To achieve a proper prefix prompt that can steer the LLM to decode the MLC without changing the LLM's parameters, we adopt the prefix-tuning (Li & Liang, 2021) method to only train a set of virtual prefix tokens as prompts to the LLM. In particular, we concat the virtual prefix and the MLC from the Q-Conformer as input to the subsequence frozen LLM. Please refer to Appendix C.3 for more details on prefix-tuning.

**Speculative Augmentation** (SA)  Despite the use of the lightweight prefix-tuning method, the size and diversity of training samples are still lacking. This is because while the Q-Conformer learns to extract task-specific context, it also learns to ignore task-irrelevant information. This would be a well-anticipated perk for an EEG encoder if we choose to directly decode language output from the EEG encoder. However, it also significantly reduces the diversity of training samples, making the learning of a good prefix difficult.

Our BELT-2 framework solves this issues by proposing the SA method to sample MLC from a total of $K+1$ Q-Conformer checkpoints to provide more diverse prefix-tuning samples. In particular, we randomly sample $K$ model checkpoints other than the best-performing checkpoint to produce MLC for the prefix-tuning. During the forward process, a speculative ratio $r$ is defined to determine whether to use best checkpoint or one of the $K$ suboptimal checkpoints. To reduce the cost of memory, we cache the output MLCs of these $K$ model checkpoints during the training of Q-Conformer to avoid actually loading the checkpoints in the prefix-tuning stage.

In our experiment, we set $K = 15$ for a balance of performance and training costs to achieve a $6\times$ larger and more diverse training sample set for the tuning of the LLM Decoder.

## 2.4 Extending Decoding to Multi-task

**Translation:**  Our definition of the EEG-to-Text translation task follows previous works on this topic (Wang & Ji, 2022). Given the word-level EEG embedding sequence and text sentence pair $\langle \mathcal{E}, \mathcal{S} \rangle$, we maximize the probability of the decoded sentence $p(\mathcal{S}|\mathcal{E})$ produced by our model. The training objective $L_{tr}$ for the translation task could be written as follows:

$$p(\mathcal{S}|\mathcal{E}) = \prod_{l=1}^{L} p(s_l|\mathcal{E}, s_{<l}), \quad \mathcal{L}_{tr} = -\sum_{l}^{L} \log p(s_l \in \mathcal{S}) \tag{6}$$

where $L$ is the total length of the target text sequence and $s_l \in \mathcal{S}$ denotes the word tokens produced by our model.

**Summary:**  We propose the first EEG-to-text summarization task by creating a summary dataset from the Zuco datasets. Human attention lingers around keywords and pivotal concepts during reading (Ding et al., 2022). Consequently, we hypothesize that the extraction of key concepts could be a more direct way to facilitate the transmission of neural information and the understanding of a

person's intention. As such, our nuanced summarization task not only enhances our understanding of EEG data but also opens up exciting possibilities for advancing research in cognitive science.

We kickstart by constructing the prompt "*Rewrite the sentence by summarizing its main idea using $\{T\}$ words from the sentence, and keep the summarized sentence similar to the original sentence:$\{s\}$*" with $\{s\}$ being each ground truth sentence from the ZuCo dataset and attain the initial summarization targets for each sentence. We set $T = 8$ in our experiment and use the LLAMA2 model (Touvron et al., 2023) to generate the initial summarization targets. Afterwards, manual inspection and rectification are carried out to improve the dataset's reliability and informativeness. The word-level EEG embedding sequence and summary pair are denoted by $\langle \mathcal{E}, \hat{\mathcal{S}} \rangle$. To extend the Q-Conformer for summarization task, a new query prompt for summarization will be added. The training objective for generating summaries is similar to Equation 6, with the sole alteration being the substitution of $\mathcal{S}$ with $\hat{\mathcal{S}}$. For multi-task training, we train all tasks simultaneously by randomly sampling tasks for each update iteration.

**Sentiment Classification:** We could further extend the Q-conformer to perform the sentiment classification task by adding another query prompt for the Q-Conformer and using the last output token from the Q-conformer as the CLS token. In particular, we use the EEG-sentiment label pair $\langle \mathcal{E}, c \rangle$. Unlike Wang & Ji (2022), we don't need to use external sentiment classification datasets or learn an additional classifier. The training objective for sentiment classification is as follows:

$$\mathcal{L}_{st} = - \sum_{i=1}^{|C|} c_i \log p(\hat{c}|\mathcal{E}_i), \quad (7)$$

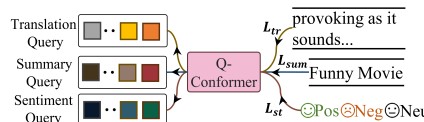

Figure 5: For multi-task training, we train three tasks simultaneously by randomly sampling tasks for each training iteration. Each task-specific query prompt learns to provide task-specific instructions by training on the corresponding task-specific objective function.

, where $|C|$ is the number of the sentiment categories and $\hat{c}$ is the sentiment prediction.

# 3 EXPERIMENT AND RESULTS

## 3.1 EXPERIMENT SETUP AND IMPLEMENTATION DETAILS

We use the ZuCo datasets (Hollenstein et al., 2018; 2019) for the training and evaluation of the proposed BELT-2 framework. The ZuCo datasets contain EEG data recorded during natural reading tasks with eye-tracking data for word-level EEG segmentation. Reading material is collected from movie reviews (Socher et al., 2013) and Wikipedia articles. We split the dataset into train, val, and test subsets (80%,10%,10%). In this cross-sentence setting, sentences will not overlap among any two subsets. In addition, cross-subject performance is also evaluated. We evaluate translation and summary performance using the BLEU scores (Papineni et al., 2002) and ROUGE-1 scores Lin (2004). We use **P.**, **R.**, **F1**, and **Acc.** to denote precision, recall, F1-score, and accuracy respectively.

## 3.2 IMPLEMENTATION DETAILS

The code could be assessed through an anonymous link [2]. For the word-level EEG embeddings, the total length of an embedding sequence is $L = 56$ and the embedding size is $d = 840$. The discrete conformer has $8$ attention heads with the feed-forward dimension size of $2048$ and a discrete codebook with $1024$ entries with a latent size of $1024$. The number of querying tokens used for The Q-Conformer is $20$. We train the Q-Conformer with a learning rate of $5e^{-06}$ for $60$ epochs during EEG-to-language alignment learning using AdamW (Loshchilov & Hutter, 2017). For the bridging stage, we use $8$ virtual prefix and set the speculative augmentation factor $K$ to $15$ with a speculative ratio of $0.3$. We use pre-trained BART and T5 models from the huggingface platform to initialize the Q-conformer and the LLM decoder. We also conducted experiments of massive size LLAMA2 model [3] in Section 3.5. Due to the limitation of space, refer to Appendix C for more details.

---

[2] https://anonymous.4open.science/r/BELT-2-0048
[3] https://huggingface.co/meta- llama/Llama-2-7b

### 3.3 TRANSLATION PERFORMANCE

**Quantitative Results** We show quantitative results in Table 1. Compared to previous methods, e.g., EEG-to-Text (Wang & Ji, 2022), Dewave (Duan et al., 2023), and BELT-1 (Zhou et al., 2023a) When only using EEG Encoder, We observe that the introduction of BPE-level contrastive learning bootstrapped a significant improvement (row 4 compared to row 5), achieving the SOTA EEG decoding BLEU-$\{1, 2, 3, 4\}$ scores of $43.06, 25.57, 15.16$, and $9.17$, which outperform DeWave by $1.71, 1.42, 1.24$, and $0.95$. By further connecting with the LLM decoder, BELT-2 further achieves the BLEU-$\{1, 2, 3, 4\}$ scores of $52.59, 36.32, 25.21$, and $17.85$, which brings additional $9.66, 10.96$, $10.16$, and $8.76$ BLEU score improvements. The increase of the metrics is more significant for longer phrases ($+162\%$ for 4-gram and $+99\%$ for 3-gram) compared to the baseline EEG-to-Text method. Additionally, we present ablation results that analyze the influence of VQ and the BPE-CL within our model, revealing that the utilization of BPE-CL significantly contributes to the enhancement of performance. However, multitask training did not bring a significant improvement to the translation result, which is elaborated in the Appendix F.

Table 1: Quantitative Results on Brain-to-Language Translation on the ZuCo Datasets.

| Model | Vector Quantizer | BPE-CL | Enable Multi-Task | Prefix Tuning | BLEU-N (%) | | | | ROUGE-1 (%) | | |
|---|---|---|---|---|---|---|---|---|---|---|---|
| | | | | | N=1 | N=2 | N=3 | N=4 | R. | P. | F1 |
| EEG-to-Text | × | × | × | × | 40.12 | 23.18 | 12.61 | 6.80 | 28.8 | 31.7 | 30.1 |
| Dewave | √ | × | × | × | 43.35 | 24.15 | 13.92 | 8.22 | 28.82 | 33.71 | 30.69 |
| BELT-1 | √ | × | × | × | 42.31 | 25.26 | 14.81 | 8.73 | 29.86 | 36.06 | 32.57 |
| **BELT-2** | √ | √ | √ | × | **43.06** | **25.57** | **15.05** | **9.09** | **30.28** | **34.12** | **31.99** |
| **BELT-2+LLM(T5)** | √ | √ | √ | √ | **52.38** | **36.28** | **25.28** | **17.95** | **36.08** | **39.47** | **37.59** |
| BELT-2 Ablations | | | | | | | | | | | |
| BELT-2 | √ | × | √ | × | 41.57 | 24.02 | 13.80 | 8.06 | 29.35 | 32.46 | 30.74 |
| BELT-2 | × | √ | √ | × | 41.90 | 24.57 | 14.2 | 8.28 | 29.60 | 34.03 | 31.54 |

Table 2: Qualitative results on unseen EEG signals. The **bold** denotes an exact match between the ground truth and our prediction. underline denotes a fuzzy match with similar semantic meanings.

| (1) | Target | **He** is a prominent **member of the** Bush **family**, the **younger brother of President George** W. **Bush and the** second son of former **President George H. W. Bush** and Barbara Bush. |
|---|---|---|
| | Others | was a former member of the American **family**, and first **brother of President George W. Bush.** the father **son** of President **President George H. W. Bush**. his Bush. |
| | Ours | **He** was great member **member of the** American **family**, and **younger brother of President George** H. **Bush and the** younger cousin of President **President George H. W. Bush**. the Bush. |
| (2) | Target | **Adolf** Otto Reinhold Windaus (December 25, 1876 - June 9, 1959) **was a** significant **German** chemist. |
| | Others | rian Hitler,hardt,eren18 18, 1885 – January 3, 18) **was a German** figure- and |
| | Ours | **Adolf** Hitlero vonhard voner (J 15, 1875 - January 15, 1945) **was a German** German industrialpacist |
| (3) | Target | **It just doesn't** have much else... especially **in** a moral **sense**. |
| | Others | was so't work the to to and not the country **sense**. |
| | Ours | **It just doesn't** work the of going except **in** the a way **sense**. |
| (4) | Target | **He was reelected twice**, but had a mixed **voting record**, often diverging **from President Harry S. Truman and the** rest **of the Democratic Party**. |
| | Others | **was** a- in, in never to less record **record**. and losingting from his Reagan **Truman**. Truman's his Republican of the Republican **Party**. |
| | Ours | **He was reelected twice**, but had voting **record**. and losingging **from President Harry S. Truman and the** other **of the Democratic Party**. |
| (5) | Target | Following the 1980 **presidential election**, Bush **and his family moved to Miami**-Dade County, Florida. |
| | Others | the deaths **election**, the was his wife **moved to** California, **Dade County, Florida.** |
| | Ours | After his election **presidential election**, Reagan **and his family moved to Miami**,Dade County, Florida. |

**Cross-Subject Results** As cross-subject performance is of vital importance for practical usage, we further report translation performance in cross-subject settings where we leave one subject out for evaluation and train the model using other subjects. Figure 6 shows the cross-subject translation performance for a total of 10 subjects compared to the cross-sentence result we achieved in the cross-sentence setting (Table 1). The radar charts in Figure 6 denote the performance is stable across different subjects with subjects achieving BLEU-1 scores ranging from $48.04$ to $51.41$.

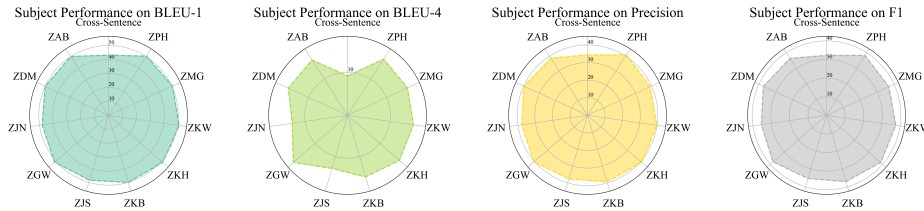

Figure 6: The cross-subjects perfromance for translation task.

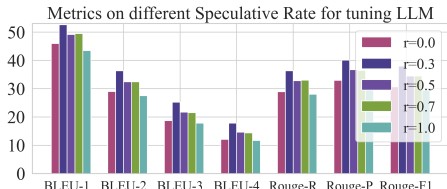

Figure 7: Ablation on speculative ratio.

| Model | BLEU (%) | | Rouge-1 | | |
|---|---|---|---|---|---|
| | N=1 | N=3 | P. | R. | F1 |
| EEG-to-Text | 25.14 | 0 | 10.37 | 7.30 | 8.49 |
| BELT-2 *w/o Pretrained* | 26.87 | 2.08 | 9.84 | 11.06 | 10.34 |
| **BELT-2 *w/ Pretrained*** | **31.17** | **5.09** | **12.73** | **13.26** | **12.91** |

Table 3: Quantitative Results of Summary Task

**Qualitative Evaluation** We showcase the generated text alongside the established approach from Wang & Ji (2022) in Table 2. We observe that BELT-2 generates more fluent sentences with greater grammatical coherence. Notably, our model adeptly captures subject-predicate relationships while other methods miss the subject and predicate. This is demonstrated by the accurate decoding of phrases like "*He was*" vs. "*He is*", "*It just doesn't work*" vs. "*It just doesn't have*". Furthermore, for sentence structures involving quoted dates, such as "*(January 15, 1875 - January 15, 1945)*" vs. "*(December 25, 1876 - June 9, 1959)*", were also consistently deciphered.

## 3.4 MULTI-TASK PERFORMANCE

**Sentiment Classification** As shown in Table 4, previous works need to train an LLM classifier using an external Stanford Sentiment Treebank dataset (around 11,000 sentences) (Socher et al., 2013) and a new EEG encoder due to poor performance when training directly on the ZoCo dataset (Row 1-3). In contrast, an EEG encoder incorporating external classifiers (row 4-7) demonstrated improved performance (Wang & Ji, 2022). Our proposed Q-Conformer Encoder, achieve the state-of-the-art sentiment classification accuracy of $74.62\%$ on the ZuCo dataset. We also observe that our method could effectively leverage pretrained knowledge from the translation task to improve performance (row 8-9).

**Summarization** We compare the summarization performance of the BELT-2 model with the EEG-to-Text model as the baseline. As shown in Table 3, the EEG-to-Text struggles to generate summarization while the proposed BELT-2 model exhibited better generative capacity, especially in longer phrases. Compared to using a newly initialized encoder (row 2), our BELT-2 exhibits a remarkable capacity to utilize the pretrained knowledge to increase the performance for the summarization task (row 3). Generally, it attains the BLEU-$\{1, 2, 3, 4\}$ scores of $31.17, 15.7, 8.91, 5.09$, outperforming the baseline method.

## 3.5 ABLATION STUDY

**Bridging Q-Conformer Encoder with different LLMs** Table 1 shows the result of bridging our Q-Conformer encoder with the T5 (Raffel et al., 2020). In Table 5, we conduct a comprehensive investigation of bridging LLM decoders with the Q-Conformer model, including the LLAMA2, T5, and the PEGASUS (Zhang et al., 2020) models. Results show that T5 LLMs consistently outperform other variants and boost the decoding performance. We attribute this superiority to T5's denoising training objectives. However, the sheer scale of the LLM decoder does not necessarily lead to enhanced decoding performance. For example, PEGASUS and LLAMA2 did not yield much improvement in the translation performance.

Table 4: Quantitative Results of Sentiment Classification

| EEG Encoder | Additional CLS Model | Additional Dataset | Acc. | P. | R. | F1 |
|---|---|---|---|---|---|---|
| MLP | None | None | 31.8 | 32.8 | 33.6 | 27.5 |
| Bi-LSTM | None | None | 30.9 | 27.5 | 33.6 | 17.4 |
| Transformer | BERT | None | 36.6 | 23.7 | 34.5 | 27.2 |
| EEG2Text | BART | SST | 55.30 | 62.40 | 56.50 | 55.60 |
| BELT-1 | BART | SST | 65.13 | 63.67 | 63.34 | 62.45 |
| BELT-1 | Albertv2 | SST | 60.09 | 61.63 | 60.03 | 59.56 |
| BELT-1 | XLNet | SST | 67.32 | 66.55 | 65.71 | 65.02 |
| BELT-2 *w/o Pretrained* | None | None | 59.74 | 57.67 | 57.63 | 57.11 |
| **BELT-2 *w/ Pretrained*** | **None** | **None** | **74.62** | **75.34** | **73.84** | **73.31** |

Table 5: Ablation study of bridging Q-Conformer Encoder with different LLMs

| LLM | Type | BLEU-N (%) | | | | ROUGE-1 (%) | | |
|---|---|---|---|---|---|---|---|---|
| | | N=1 | N=2 | N=3 | N=4 | P. | R. | F1 |
| LLAMA2 | 7B | 21.40 | 6.96 | 3.38 | 2.21 | 12.23 | 13.20 | 12.61 |
| PEGASUS | google/pegasus-x-base | 37.67 | 18.90 | 9.68 | 5.21 | 26.43 | 31.06 | 28.38 |
| | google/pegasus-xsum | 40.82 | 23.70 | 13.39 | 7.61 | 30.25 | 33.94 | 31.86 |
| T5 | t5-small | 51.02 | 33.44 | 22.41 | 15.42 | 34.91 | 37.80 | 36.15 |
| | t5-base | 51.36 | 33.75 | 22.74 | 15.63 | 35.09 | 38.19 | 36.41 |
| | **t5-large** | **52.59** | **36.32** | **25.21** | **17.85** | **36.32** | **40.10** | **38.00** |
| | google/flan-t5-base | 50.01 | 33.09 | 21.77 | 14.49 | 32.97 | 36.64 | 34.54 |
| | google/flan-t5-large | 49.85 | 33.08 | 22.07 | 14.84 | 33.11 | 36.61 | 34.59 |

**Speculative Augmentation**   We further conduct ablation experiments on the effect of different speculative ratios in Figure 7. We observe that the introduction of speculative augmentation at $r = 0.3$ has a significantly better impact on the decoding performance across all evaluated metrics.

## LIMITATIONS

While BELT-2 achieved remarkable translation improvements by combining Q-Conformer with LLMs, it is worth noting that the accuracy still lags behind traditional language-to-language translation. Also, it is noted that the experiments were conducted on publicly available neural reading datasets with the help of eye-tracking markers. As a result, BELT-2 has not realized everyday communication such as 'silent speech' or 'reading mind'. The vision of communication or controlling devices directly from brain dynamics remains a challenging task for follow-up research.

## 4   CONCLUSION

This paper introduces BELT-2, a pioneering EEG-language learning framework for bridging brain signals to LLMs. Our framework achieves EEG-to-language alignment by incorporating the novel BPE-CL objective and proposed an effective method for bridging a frozen Q-Conformer EEG Encoder and a frozen LLM to leverage their generative capacity. The multi-task extendibility of the Q-Conformer also establishes BELT-2 as the first work to achieve a multi-task decoding model in EEG research. Extensive experiments were conducted to evaluate the performance of BELT-2 quantitatively and qualitatively. Especially, this work provides the first study investigating the feasibility of using frozen pretrained LLM to process EEG contexts exampled by a wide range of LLMs. Our experimental result shows that the BELT-2 framework represents a significant step forward in integrating human brain signals with LLMs, opening up exciting new avenues for research and development in cognitive neuroscience and brain-computer interfaces. We hope that this work will inspire further exploration and innovation in this exciting and rapidly evolving field.

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
