## Supplementary Material for BELT-2: Bootstrapping EEG-to-Language representation alignment for multi-task brain decoding

## A Related Works

**EEG decoding**    Prior brain studies demonstrated the potential to decode speech (Anumanchipalli et al., 2019) and language signals (Anumanchipalli et al., 2019) from the human brain using invasive neuro-sensors, but the risks make it impractical for most people. More recently, a surge of efforts was made to extract rich information from noninvasive brain signals through advanced representation learning techniques, opening the door to a wide array of innovative tasks based on brain signals, such as image reconstruction (Singh et al., 2023) and movement prediction (Zhou et al., 2023b). Nonetheless, Many of these efforts have limitations, including vocabulary size and decoding performance, hindering their suitability for complex practical scenarios. Our work focuses on open-vocabulary sentence decoding from noninvasive brain signals with fluent decoding performance and versatile multi-task adaptability, making it a promising solution for a diverse range of applications.

**EEG-Language representation alignment**    A crucial step for most cross-modality tasks is the acquisition of aligned multi-modal representations (Liu et al., 2023a; Mokady et al., 2021; Rombach et al., 2022). Achieving this involves an alignment step following the acquisition of unimodality pretrained models (Li et al., 2023). Yet, the formidable challenge persists due to the limited scale and sparse EEG dataset annotations, as we strive to create a semantically coherent and universally adaptable EEG encoder, akin to visual counterparts (Dosovitskiy et al., 2020; Radford et al., 2021).

Diverging from the conventional fully-supervised paradigm, infusing natural language supervision enriches non-language modalities representation with semantics and zero-shot generalization (Desai & Johnson, 2021). Previous studies in unimodal vision tasks show that a large vision encoder, trained directly with language supervision, can match performance compared to learning from massive datasets (Joulin et al., 2016). Recent works incorporating language-guided learning also support the value of additional semantics for non-language representation generalization (Wang et al., 2023; Elizalde et al., 2023). Inspired by their successes, our work endeavors to bootstrap the learning of an Encoder that aligns EEG and language representation through natural language supervision.

## B Mathematical symbols used in this paper

In Table 6 we show a list of mathematical symbols used in this paper.

Table 6: List of mathematical symbols used in this paper

| Symbol | Description | Symbol | Description |
|---|---|---|---|
| $\langle \mathcal{E}, \mathcal{S} \rangle$ | Word-level EEG embedding sequence and text sentence pair | $\langle \mathcal{E}, c \rangle$ | Word-level EEG embedding sequence and sentiment label pair |
| $\langle \mathcal{E}, \hat{\mathcal{S}} \rangle$ | Word-level EEG embedding sequence and text summary pair | $\mathbf{w} \in \mathcal{W}$ | BPE text token's embeddings |
| | | $\mathbf{e} \in \mathcal{E}$ | EEG embedding vector |
| $c \in \mathcal{C}$ | Sentiment label | $\mathbf{v} \in \mathcal{V}$ | Discrete codebook embeddings |

## C Implementation Details

### C.1 Implementation Details for the Q-Conformer

The Q-Conformer is implemented using the configuration detailed in Table 7. The detailed structures for the convolution module are shown in Table 8. We use the same Conformer block for the encoder and decoder, each with 2 Conformer blocks. We trained All models are trained on Nvidia A40 GPUs.

Table 7: Detailed configuration of the conformer block

| Layer | Hidden Size | Activation Function | Number of Heads |
|---|---|---|---|
| Layer Norm | 840 | - | - |
| Feed Forward Module | 840 | GELU | - |
| LayerNorm | 840 | - | - |
| Multi-Head Self Attention | 840 | - | 8 |
| Convolution Module | 840 | - | - |
| Layer Norm | 840 | - | - |
| Feed Forward Module | 840 | GELU | - |
| LayerNorm | 840 | - | - |

Table 8: Detailed configuration of the convolution module

| Layer | Kerrnel | Stride | In Channel | Out Channel |
|---|---|---|---|---|
| Layer Norm | - | - | 840 | 840 |
| Pointwise Convolution | 1 | 1 | 840 | $2 \times 840$ |
| Depthwise Convolution | 31 | 1 | 840 | 840 |
| Batch Norm | - | - | 840 | 840 |
| Pointwise Convolution | 1 | 1 | 840 | 840 |
| Dropout | - | - | - | - |

## C.2 TRAINING DETAILS FOR EEG-TO-LANGUAGE ALIGNMENT LEARNING

To train the Q-Conformer during the EEG-to-language alignment learning, we use a weighted summation of all the following loss terms:

$$\mathcal{L} = \lambda_1 \mathcal{L}_{vq} + \lambda_2 \mathcal{L}_{bpe} + \lambda_4 \mathcal{L}_{elm} + \lambda_3 \mathcal{L}_{neg}, \tag{8}$$

$\lambda_1$ to $\lambda_4$ are coefficients for each loss term. We set $\lambda_1$ to $\lambda_4$ as $[1, 10, 10, 0.001]$. The main reason for such a setting is the aim to prioritize the learning of achieving EEG-to-language alignment and the training of the query prompt specific to the ELM task. To avoid collapse in training, we implemented the gradient normalization method to normalize the scale of the loss function and stabilize the training process.

## C.3 TRAINING VIRTUAL PREFIX FOR BRIDGING Q-CONFORMER AND LLM

The prefix-tuning method used in our paper closely follows the implementation in Li & Liang (2021), the objective function ($\mathcal{L}_{bridge}$) is defined as a modified loss function tailored to guide the selective of continuous virtual prefix prompts. We use $\theta$ to denote the matrix that stores the virtual prefix. Using the machine translation loss $\mathcal{L}_{tr}$ as an example, the objective function can be expressed as:

$$\mathcal{L}(\theta_{\text{bridge}}) = \mathcal{L}_{tr}(\hat{\mathcal{S}}, \mathcal{S}) \tag{9}$$

In this example, the prefix prompts to learn properly describe the EEG-to-Langugage translation task to the subsequence frozen LLM, utilizing the generation capacity of the LLM models to improve translation performance.

## C.4 TRAINING DETAILS FOR MULTI-TASK LEARNING

To extend our model to multi-task decoding, we simultaneously train the model in three EEG decoding tasks including translation, summary, and sentiment classification task. We randomly sample a task for each batch during the training epochs. The loss function for translation task $\mathcal{L}_{tr}$ and sentiment classification tasks $\mathcal{L}_{st}$ are illustrated in Equation 6 and Equation 7 respectively.

For learning the summary task, the loss function could be written as follows:

$$\mathcal{L}_{sum} = -\sum_{l}^{|\hat{\mathcal{S}}|} \log p(s_l \in \hat{\mathcal{S}}) \tag{10}$$

, where $p(s_l)$ denotes a model predicting the word token for the next location. The final multi-task objective $\mathcal{L}$ is written as follows:

$$\mathcal{L}_{mt} = \mathcal{L}_{tr} + \mathcal{L}_{sum} + \mathcal{L}_{st} \tag{11}$$

## D    IMPROVED Q-CONFORMER EEG ENCODER

We observed a noteworthy trend when utilizing a relatively larger learning rate of $1e-4$, as opposed to the optimal learning rate of $5e-6$ for the top-performing Q-Conformer Encoder, as indicated in Figure 8. This variance in learning rates led to a remarkable performance by the Q-Conformer Encoder on the training dataset, resulting in notably high BLEU Scores. Specifically, the BLEU-1 and BLEU-4 scores soared to remarkable levels, reaching 93.03 and 92.69 respectively. In stark contrast, the EEG-to-Text baseline method significantly lagged behind, registering only BLEU-1, 4 scores of 38.98 and 6.82 during our replicated training, highlighting the superior EEG encoding capabilities of the Q-Conformer Encoder.

It's also worth noting that the BLEU-1 performance of the Q-Conformer encoder experienced a decline from 42.43 to 35.48 during the testing phase, we interpret this as a minor setback. Such a reduction in performance can often be attributed to the challenges of generalization, which frequently happen in the context of training on a relatively small dataset.

Furthermore, it's worth highlighting that within this setting, the Q-Conformer still achieved a testing BLEU-4 score of 9.3, surpassing the baseline EEG-to-Text method's training set BLEU-4 score. This outcome serves as a compelling testament to the enhanced encoding capacity conferred by our Q-Conformer Encoder.

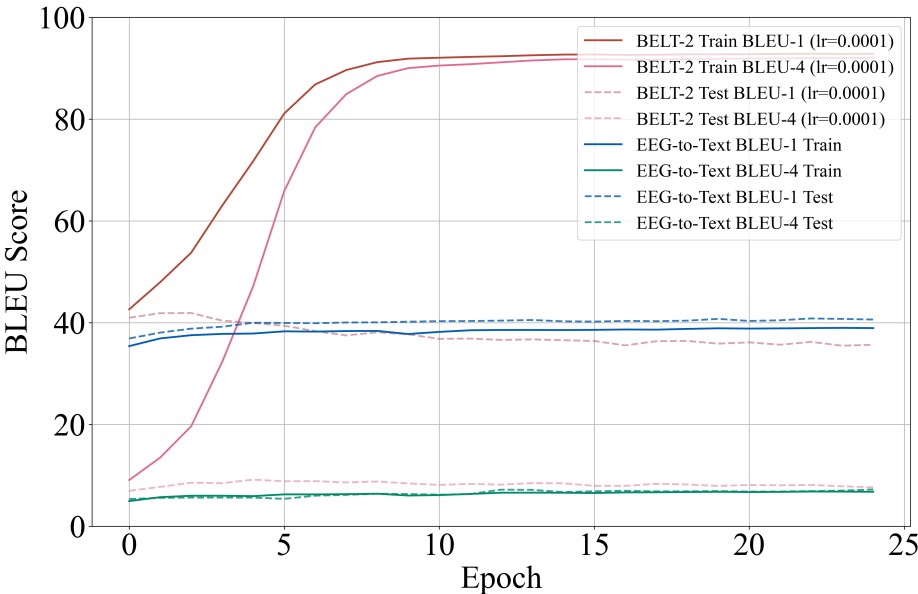

Figure 8: EEG encoder performance comparison

## E    COMPARISON WITH BELT-2 WITHOUT BPE-LEVEL CONTRASIVE LEARNING

In Figure 9(a) and Figure 9(b), we present a comprehensive comparison of the learning curves and BLEU-1 curve of the baseline EEG-to-Text model (Cruttenden, 2014), the Q-Conformer encoder without applying the BPE-level contrastive learning (BELT-2 w/o BPE-CT) and the Q-Conformer

encoder with BPE-level contrastive learning (BELT-2 w/ BPE-CT)g. The visualized learning curves include the BLEU-1 score and loss values for 30 epochs on the test split. Comparing the EEG-to-Text model and the BELT-2 model, it's evident that BELT-2 offers a significant reduction in loss values with or without BPE-level contrastive learning, indicating the proposed model architecture is more efficient in capturing EEG patterns. However, a notable observation arises after epoch 8. Without the BPE-contrastive learning (orange curves), the BLEU-1 score fluctuates and drops significantly. On the contrary, the introduction of BPE-level loss helps stabilize the model's performance, particularly on unseen EEG data. This highlights the substantial enhancement brought about by our proposed BPE-contrastive learning framework.

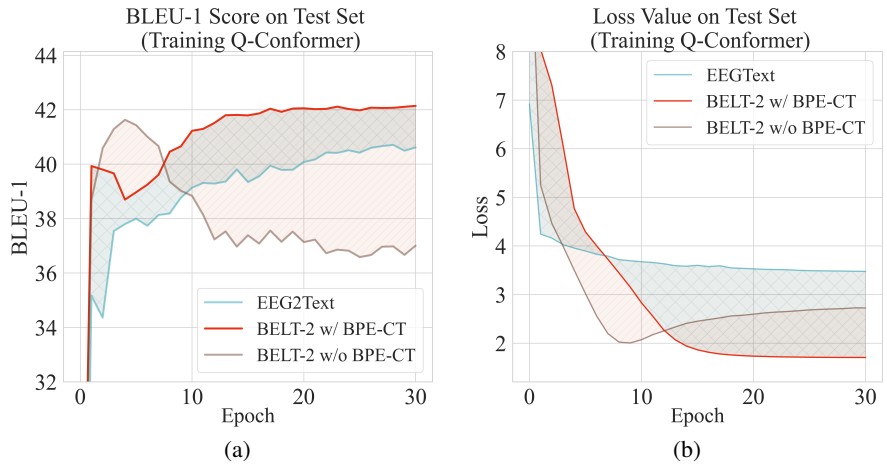

Figure 9: Ablation Study on Different Settings

## F    MULTI-TASK TRAINING RESULTS

We show the performance of translation, summary, and sentiment classification on the test set during the multitask training learning phase of BELT-2 in Table 10. In Table 10(a), we can observe that without the use of pretrained weights, all tasks are learned from scratch. In this case, the translation BLEU-1 score starts from 4.06 BLEU-1 score and rises to only reaches 41.47 and the summarization BLEU-1 score reaches 28.72. Also, the sentiment classification accuracy gradually increased to 59%. However, the use of Q-Conformer pretrained on translation tasks could improve the training stability and performance of both the sentiment classification task and the summarization task. Due to the pretrained weights, we observed that in Table 10(b), the BLEU-1 score of the summarization performance and sentiment achieved 23.0 BLEU-1 score after the first training epoch. Then continued to increase to 31.17. The accuracy for sentiment classification also reaches 79.86% at its peak and stabilizes at around 74%. However, the performance of the translation task slightly decreased. This is an expected phenomenon in multi-task training. Nonetheless, this ethernet still shows the multi-task learning capacity and extensibility of our BELT-2 framework.

## G    GENERATED SUMMARIZATION RESULTS

We created the summarization dataset with the prompt "Rewrite the sentence by summarizing its main idea using 8 words from the sentence and keep the summarized sentence similar to the original sentence: {$s$}" where and {$s$} is the original sentence from the dataset. Table 9 showcases summary and prediction samples generated by the BELT-2 model. We could see those summary ground truths cover the key ideas of the original sentence and are within the maximum summarization word limit. On the training set, our BELT-2 model could learn and precisely generate a summary of the EEG signal, such as "film with twists" vs. "film with twists.". However, this summarization capacity did not generalize well on unseen test and validation data. We consider the lack of training data as one of the major reasons for this problem. Another reason is that our current model lacks higher-level

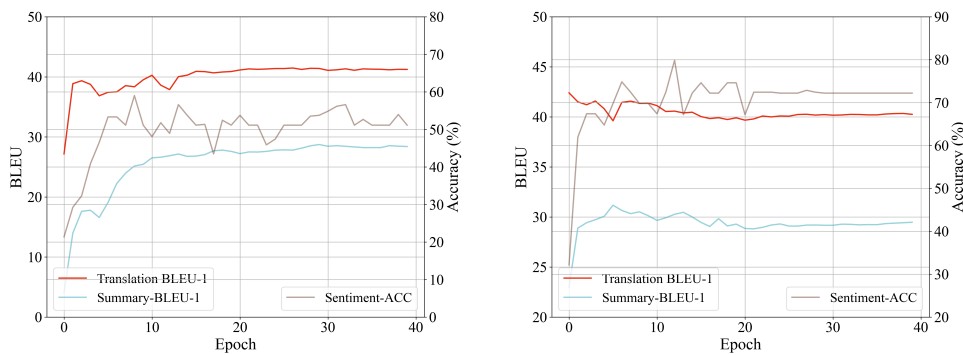

(a) Multi-task training **without** pretrained Q-Comformer Weights

(b) Multi-task training **with** pretrained Q-Comformer Weights

Figure 10: Ablation study on multitask learning and effect of our pretrained weights

skill that requires additional reasoning and abstraction skills beyond the mere translation of the brain signal, which leaves room for future improvements.

## H    ABLATION EXPERIMENTS ON HYPER-PARAMETERS

We conducted an ablation study on different hyper-parameters including the learning rate, batch size, frequency of the inserted cross-attention layer in the context layer of the Q-Conformer, and the number of querying prompts. The evaluation metrics can be found in Figure 11. We observe that the introduction of BPE-contrastive learning consistently improves training stability and model performance in different hyper-parameter settings. This result shows that the learning performance of BELT-2's EEG encoder is not easily affected by the change of training parameters and is relatively easy to reproduce.

## I    AUGMENTATION EFFECT OF SPECULATIVE AUGMENTATION

The limitation of unique sentence from the training dataset also limits the diversity of the MLC context outputed by the Q-Conformer. The training set we used in our cross-sentence setting contains only 790 unique sentences as target for prefix-tuning when bridging Q-Conformer and LLM. For the Q-Conformer, predicts around 900 uniques MLC throughout the training dataset. This lack of training inputs makes the training for a good virtual prefix difficult. To solve this problem, our speculative augmentation method reuse cached MLC from the training stage of Q-Coformer. When using MLC from $K = 15$ checkpoints, we achieve a total of $5107$ samples for prefix-tuning.

## J    EXTENSIVE EXAMPLES OF GENERATED TRANSLATION OUTPUTS

We provide extensive translation outputs from our BELT-2 model compared with the baseline EEG-to-Text model and the ground truth in Table 10. It shows that for some samples, the BELT-2 model still has insufficient performance, which indicates room for future improvements.

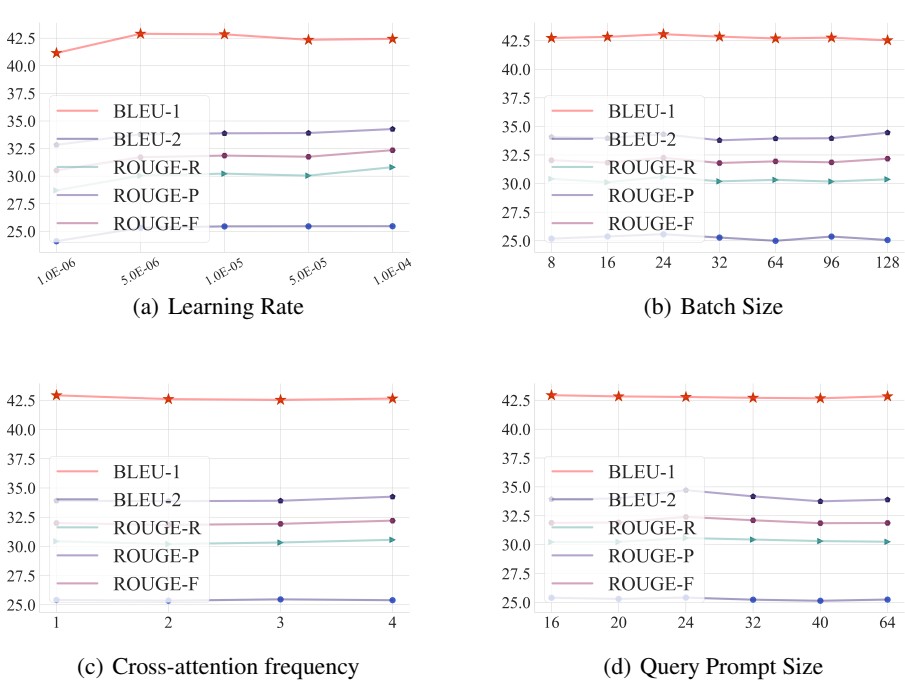

(a) Learning Rate

(b) Batch Size

(c) Cross-attention frequency

(d) Query Prompt Size

Figure 11: Ablation study on hyper-parameters.

Table 9: Summarization examples and generated results on the train set. The **bold** denotes an exact match between the ground truth and our prediction. underline denotes a fuzzy match with similar semantic meanings.

| | | Training |
|---|---|---|
| (1) | Sentence | Beautifully crafted, engaging filmmaking that should attract upscale audiences hungry for quality and a nostalgic, twisty yarn that will keep them guessing. |
| | Summary GT | High-**quality film with twists.** |
| | Prediction | **-quality film with twists.** |
| (2) | Sentence | Slow, silly and unintentionally hilarious. |
| | Summary GT | Silly, **slow** comedy. |
| | Prediction | inger, **slow** movie. |
| (3) | Sentence | The movie is for fans who can't stop loving anime, and the fanatical excess built into it. |
| | Summary GT | Anime **fans will love** excessive movie. |
| | Prediction | imated **fans will love** this gore. |
| (4) | Sentence | But here's the real damn: It isn't funny, either. |
| | Summary GT | Funny, but **not really.** |
| | Prediction | unny, smart **not really.** |
| (5) | Sentence | Everything was as superficial as the forced New Jersey lowbrow accent Uma had. |
| | Summary GT | Uma's **accent was fake.** |
| | Prediction | ma's **accent was fake.** |
| (6) | Sentence | Feels like nothing quite so much as a middle-aged moviemaker's attempt to surround himself with beautiful, half-naked women. |
| | Summary GT | Filmmaker surrounds **himself with beautiful women.** |
| | Prediction | mmakers imagined **himself with beautiful women.** |
| (7) | Sentence | He died in Springport, New York in 1815. |
| | Summary GT | Man **passed away in Springport.** |
| | Prediction | **passed away in Springport.** |
| | | Test and Validataion |
| (1) | Sentence | A richly imagined and admirably mature work from a gifted director who definitely has something on his mind. |
| | Summary GT | Director's mature work reflects deep thoughts. |
| | Prediction | 's debut film. his empathy. |
| (2) | Sentence | An amateurish, quasi-improvised acting exercise shot on ugly digital video. |
| | Summary GT | **Ugly** video showcases poor **acting.** |
| | Prediction | ma,, **ugly acting.** |
| (3) | Sentence | Warm Water Under a Red Bridge is a quirky and poignant Japanese film that explores the fascinating connections between women, water, nature, and sexuality. |
| | Summary GT | Japanese film explores women, water, nature, sexuality poignantly. |
| | Prediction | actor, themes's love, and. love.eticsancy. |
| (4) | Sentence | It just doesn't have much else... especially in a moral sense. |
| | Summary GT | Limited moral **compass** |
| | Prediction | role **compass**. |
| (5) | Sentence | It's solid and affecting and exactly as thought-provoking as it should be. |
| | Summary GT | Thought-**provoking** and solid. |
| | Prediction | inful**provoking** film funny. |
| (6) | Sentence | The art direction is often exquisite, and the anthropomorphic animal characters are beautifully realized through clever makeup design, leaving one to hope that the eventual DVD release will offer subtitles and the original Italian-language soundtrack. |
| | Summary GT | Beautiful animal characters, DVD subtitles. |
| | Prediction | iful, inter. funny experience. |

Table 10: Extensive examples of generated translation outputs from unseen EEG signals in the test set. The **bold** denotes an exact match while underline denotes a fuzzy match with similar semantic meanings.

| (1) | Target | **It's not a** particularly good film, **but** neither **is it** a monsterous **one**. |
| | Others | was a a bad good story, but it is it bad bad. one. |
| | Ours | **It's not a** bad bad movie, but it **is it** kinda good bad **one**. |
| (2) | Target | It's solid and affecting and exactly as thought-**provoking as** it should **be**. |
| | Others | was a, it, it what it.provoking as it is be. |
| | Ours | **It's**, believable, is what -**provoking as** the sounds **be**. |
| (3) | Target | Co-writer/director Jonathan Parker's attempts to fashion a Brazil-like, hyper-real satire fall dreadfully short. |
| | Others | operfounder of**director** of Dem is novel to make a film-themed film but-realistic of flatfully short of |
| | Ours | Theenstarrings**director** John Dem hass films to make a new-style film -realisticromre are flatareadfully flat. |
| (4) | Target | **After** World **War II,** Kennedy entered politics (partly to fill **the void** of his popular brother, Joseph P. **Kennedy**, Jr., on whom his family had pinned many **of their hopes** but who was killed in the war). |
| | Others | the War II, the was the andasly as serve the void left a father father , John Kennedy. **Kennedy**, who.) who the he father had been their of his **hopes**). never was never in the war). |
| | Ours | **After the War II,** became politics,andly to fulfill **the void** left his father father, John **Kennedy**. Kennedy, who.,who the Kennedy family had placedbased their **of their hopes**). had had in Battle. |
| (5) | Target | **It's** solid and affecting and exactly as thought-**provoking** as it should be. |
| | Others | was a, it, it what it.outoking as the sounds be. |
| | Ours | **It's**, logical, is what -**provoking** as the sounds be. |
| (6) | Target | **Too much** of this well-acted but dangerously slow thriller feels like a preamble to a bigger, more complicated story, one that never materializes. |
| | Others | bad of a is-known, not over- is like a film-ble to a more, more dramatic story. which that will quiteizes. |
| | Ours | **Too much** drama is-made, unly un-. like a -ble to a much, more serious,. one that' quiteizes. |
| (7) | Target | In 1923 **he was awarded the** inaugural Bôcher Memorial**Prize** by the American Mathematical **Society**. |
| | Others | the, married born the Nobel Pulitzericentne **Prize** Medal for the French Academyical **Society**. |
| | Ours | In 1815,**he was awarded the** Pulécher Prize **Prize**, the Royal Academyematical **Society**. |
| (8) | Target | **He later became** an educator, teaching music theory **at the University of** the District of Columbia; he was also director of the District of **Columbia** Music Center jazz workshop band. |
| | Others | was **became** a actor and and at and and the University of California Arts of Columbia. and also also a of the University of **Columbia**'s School. department.. |
| | Ours | **He later became** associate at and at at **at the University of** California West of Columbia and and he also of the English' **Columbia**' Department. department. |
| (9) | Target | **Fans of the** TV series **will be** disappointed, and everyone else will be slightly bored. |
| | Others | of the film show " remember familiar to however the will will be happy amused. |
| | Ours | **Fans of the** movie series **will be**, as the who will be left disappointed. |