# OpenReview forum: "BELT-2: Bootstrapping EEG-to-Language representation alignment for multi-task brain decoding"
_ICLR.cc/2024/Conference — Submitted to ICLR 2024_

### Official Review · Reviewer_6hGH · 2023-10-25

**Soundness:** 2 fair
**Presentation:** 3 good
**Contribution:** 2 fair
**Rating:** 1
**Confidence:** 5

**Summary:**

This paper proposed a model named BELT-2 for multi-task EEG-to-Language decoding.  In particular, a discrete conformed is used to convert EEG  into EEG embeddings. Combined with a soft query prompt, the Querying Discrete Conformer (Q-Conformer) enables the multi-task mechanism. The EEG embedding is aligned with BPE tokens and fed to LLM such as T5. The experiment is conducted on an open-source ZuCo dataset to show the effectiveness of BELT-2.

**Strengths:**

The idea of utilizing the powerful representation ability of LLM for EEG decoding is interesting.

**Weaknesses:**

* a) Many of the training details are unclear, which makes it very hard to understand the working mechanism of the proposed BELT-2. I will elaborate in the question section.

* b) The training of the Q-Conformer is based on seven different loss terms λ1(L_cm + L_cb) + λ2 * L_recon + λ3 * L_div + λ4 * L_bpe + λ5 L_neg + λ6 * L_tr with a list of balancing coefficients of [1, 1, 0.001, 10, 0.001, 10]. Training a discrete encoder (Conformer in this case) like VQ-VAE along with the downstream (translation in this case) is not common. From my experience, balancing the reconstruction quality and the code quality itself while training the VQVAE is already a hard task. Thus, I would very much like to know how this model would perform without the quantization and reconstruction. That is to remove L_cm, L_cb, L_recon, and L_div and only use the Conformer as an EEG encoder. Ideally, an ablation study on these loss terms and corresponding curves during training would help the readers understand the different components of BELT-2.

* c) The experimental setting of word-level modeling is questionable. For a regular sentiment classification task, a continuous EEG signal is used as input, yet this work only extracts EEG segments based on eye tracking. Naively speaking, sentiment-related information does not necessarily exist only when the subject is looking at words. Or else, there might be a delay of the reaction in the brain upon reading. Is this considered while aligning the word and EEG?

* d) The word-level modeling seems too artificial to fit the word embedding format to LLM. Sequence-to-sequence modeling would be more interesting and practical.

* e) Speculative Augmentation takes k=15 other copies of the Conformer, which is both computationally and memory expensive.  An ablation on K should also be provided.

**Questions:**

* a) The training details of the Q-Conformer is only provided in the appendix but nowhere to be mentioned in the main text. The authors should not use the appendix as additional pages to the paper.

* b) What is the vocabulary used during training the EEG-language alignment? Is it the vocabulary of the ZuCo dataset or the vocabulary of the LLM (T5) ?

* c) It is also not clear what loss function is used to train the multi-task query and which part of the network is updated. Is it the context transformer or the query prompts?

* d) It is not clear why BPE is adopted. What would happen if the contrastive loss is used without BPE, meaning use whole word tokens?

* e) When combining Q-Conformer with  LLM using continuous virtual prompts, how is it trained exactly?  These are not even mentioned in the appendix.

I would like to raise my rating with my concerns and questions addressed.

---

> ### Author Response · Authors · 2023-11-14
>
> ### Q 1:
> The training details of the Q-Conformer is only provided in the appendix.
> ### R 1:
> We put training details in the appendix mainly due to the page limitation. We will reorganize and put the details into the main paper in the final version.
>
> ### Q 2:
> Question about vocabulary size.
> ### R 2:
> The vocabulary in the training split of the ZuCo dataset has 2603 unique words. However, in the final translation result in the  the decoding phase, we use the vocabulary of the LLM (T5) which is 32128.
>
> ### Q 3:
> Not clear what loss function is used to train the multi-task query and which part of the network is updated.
> ### R 3:
> In the multi-task training experiment, we train 3 tasks simultaneously by randomly sampling tasks in each training step. We also found the best performance came from updating the whole Q-Conformer encoder, including the discrete conformer, the query prompt, and the context transformer.
>
> ### Q 4:
> Question about the BPE method.
> ### R 4:
> We provide an additional ablation experiment involving using word-level contrastive learning with a discrete Conformer model involving three different settings.
>
> Results are detailed in the table. The use of word-level improves the BLEU scores to an extent while the BPE-level contrastive learning leads to further improvements for all BLEU scores. This rationalizes the use of BPE-level contrastive learning over word-level contrastive learning.
> | Contrastive Learning | Bleu-1 | Bleu-2 | Bleu-3 | Bleu-4 |
> | --------------------- | ------ | ------ | ------ | ------ |
> | None                  |41.57  | 24.03  | 13.8  | 8.06    |
> | Word-level            | 42.31  | 25.26  | 14.81  | 8.73   |
> | BPE-level             | 43.06  | 25.57  | 15.16  | 9.17   |
>
> ### Q 5:
> When combining Q-Conformer with LLM using continuous virtual prompts, how is it trained exactly?
> ### R 5:
> In Section 2.4 and the introduction, we briefly introduced the training approach involving the frozen Q-Conformer and the pre-trained Language Model (LLM) decoder using the **prefix-tuning** method. While this method is commonly employed in fine-tuning LLMs.
>
> The essence of the prefix-tuning is to only tune the prefix tokens to the LLM so that the trained prefix contains enough instruction for the LLM to understand the task [1].
>
> Since the prefix-tuning method is not a primary contribution introduced by our work, we did not delve into extensive details in the main paper.
>
> [1] Prefix-tuning: Optimizing continuous prompts for generation.
>
> ### Q 6:
> Question about the training of the Q-Conformer.
> ### R 6:
> We conducted additional ablation experiments were conducted on the Q-Conformer encoder to show performances when trained with or without the VQ and contrastive components. The results indicate that VQ and BPE-Contrastive could improve the model's performance together. We will add the ablation in the final version of the paper.
>
> | VQ  | BPE-Contrastive | Bleu-1 | Bleu-2 | Bleu-3 | Bleu-4 |
> | --- | ---------- | ------ | ------ | ------ | ------ |
> | No  | No         | 40.12  | 23.18  | 12.61  | 6.8    |
> | No  | Yes        | 41.9   | 24.57  | 14.2   | 8.278  |
> | Yes | No         | 41.57  | 24.02  | 13.80  | 8.06   |
> | Yes | Yes        | 43.06  | 25.57  | 15.16  | 9.17   |
>
>
> ### Q 7:
> Question about the experimental setting of word-level modeling.
>
> ### R 7:
> There are two major reasons for using word-level tokens as input in conducting the sentiment classification task.
>
> The first reason is that we strictly follow the experimental setup in the baseline method for a fair comparison.
>
> Another reason is that we want to demonstrate the multi-task capacity of our model. This means that given the same input, our model is able to decode different information from the input EEG tokens (i.e., the translation, summary, and sentiment).
>
> ### Q 8:
> Question about word-level modeling.
>
>
> ### R 8:
> We add ablation experiments on different contrastive learning levels to answer this question as shown below.
>
> [Translation Task]
> | Contrastive learning | BLEU-1 | BLEU-2 | BLEU-3 | BLEU-4 |
> | -------------------- | ------ | ------ | ------ | ------ |
> | word-level           | 42.31  | 25.26  | 14.81  | 8.73   |
> | sequence-level       | 42.23  | 24.95  | 14.29  | 8.14   |
> | BPE-level            | 43.06  | 25.57  | 15.16  | 9.17   |
>
> We observe that the use of sequence-to-sequence modeling did not yield a significant improvement when compared to word-level modeling.
>
> ### Q 9:
> Question about Speculative Augmentation.
> ### R 9:
> The Speculative Augmentation does not incur extra memory requirements. we employ a cache system by saving copies of the intermediate layer output (last output tokens) from the training set at each epoch. The cached outputs from a total of 15 training epochs, are utilized to diversify the training data spectrum during the prefix-tuning process. Thus, we use these cached outputs instead of loading checkpoints.

---

> ### Author Response · Authors · 2023-11-21
>
> ### Extended response to Weakness a:
> To further address the problem of the clarity of the training process, we have improved the readability and organized the illustration of the learning process. The training and loss function of Q-Conformer is clarified in Section 2.2 where we summarize the EEG-to-langauge alignment learning. In this stage, we jointly optimize 3 objectives, BPE-level contrastive learning (BPE-CL), Nagative contrastive learning (NCL), and the EEG-to-language matching (ELM) with the loss function clearly presented in the section. The first two objectives are applied to the discrete EEG tokens to improve the semantic (EEG-language alignment) while keeping the distinction among different tokens. The third objective is for training a task-specific query prompt as well as training the context-transformer. For multi-tasking, we provided a clearer description in Sectopm 2.4 and Figure 5. Please refer to the current paper for more details.

---

> ### Author Response · Authors · 2023-11-23
>
> Dear Esteemed Reviewer,
>
> Thank you immensely for your invaluable insights and thoughtful queries that have significantly contributed to enhancing our work. Your feedback has been meticulously incorporated into our paper, elevating its quality and depth.
>
> We earnestly appeal to your esteemed judgment once again, considering the substantial refinements made, to reconsider the rating for our paper. Our relentless dedication has been aimed at pushing the boundaries within the realm of EEG research, aspiring to pave the way for broader applications and advancements.
>
>
> We hope our work can help improve brain research and make a positive difference in the wider research community. Your support and revised evaluation would not only recognize our efforts but also amplify the potential impact of this research, steering us collectively toward greater progress.
>
> With sincere regards,
> The Authors

---

> > ### Comment · Reviewer_6hGH · 2023-12-02
> > **Further Concerns on the Training Details of Q-Conformer**
> >
> > I raised the question in weakness b) about the training of the Q-Conformer. In the original version of the appendix, the authors stated that the training of the Q-Conformer is based  λ1(L_cm + L_cb) + λ2 * L_recon + λ3 * L_div + λ4 * L_bpe + λ5 L_neg + λ6 * L_tr with a list of balancing coefficients of [1, 1, 0.001, 10, 0.001, 10]. Without answering my question, the authors changed the number of losses used to train the Q-Conformer **to only four** in the updated version of the appendix and manuscript. The new set of balancing coefficients seems incompatible with the previous ones. I wonder how exactly this Q-Conformer is trained and which version of Q-Conformer the experimental results yielded in this paper are based on.

---

### Official Review · Reviewer_CXNw · 2023-10-31

**Soundness:** 3 good
**Presentation:** 3 good
**Contribution:** 3 good
**Rating:** 8
**Confidence:** 2

**Summary:**

The manuscript presents BELT-2, a novel multi-task model that bridges the capabilities of large language models with human brain dynamics, focusing on enhancing the encoding and decoding of EEG signals. With its BPE-level EEG-language alignment and multi-task training, BELT-2 marks a significant breakthrough in EEG decoding. The paper boasts impressive results in multiple tasks, such as EEG-to-Text Translation, EEG Sentiment Classification, and EEG-to-Text Summarization. Notably, the BELT-2 model outperforms the state-of-the-art in several benchmarks, demonstrating its effectiveness.

**Strengths:**

1. The manuscript is well composed, with a clear structure and logical flow, enhancing the reader's understanding and engagement.
2. The experiments are comprehensive. The detailed comparisons to state-of-the-art methods across various tasks, complemented by thorough ablation studies, add significant depth and robustness to the paper's findings.

**Weaknesses:**

1. Figure 3 presents EEG topography plots for both the input and output during the EEG token quantization process, leading to some ambiguity in interpretation. I would recommend the authors to elucidate this procedure in greater detail. Specifically, it would be insightful to understand whether the spatial arrangement of the EEG sensors played any role in this process.

2. The manuscript introduces BELT-2 as a progression from the prior BELT-1 model. However, the discussion and distinction between the two models are somewhat scanty, especially given their apparent similarities. It would be of immense value if the authors could elaborate on the design improvements made in BELT-2 over BELT-1. A focused discussion highlighting the specific enhancements and their contribution to the performance improvements, as showcased in Table 1 and Table 4, would add depth to the paper.

3. A few inconsistencies are observed in the formatting of the tables, which might be distracting for readers. I'd kindly suggest revisiting and refining the table presentation to ensure a consistent and polished format.

4. In Figure 4 and Section 2.4, there is a mention of utilizing the mediate layer coding as 'EEG prompts'. The concept, as presented, leaves some gaps in understanding, primarily because its introduction and visualization seem absent or not explicitly labeled in the preceding figures and method sections. It would enhance coherence and clarity if the authors could revisit Figures 2 and/or 3 and annotate the specific parts illustrating this mediate layer coding.

**Questions:**

In the section detailing the experimental setup, the authors introduced the dataset split. Was this split done on a cross-subject basis or just cross-training samples? Given the well-documented variations in EEG signals across different individuals, understanding this aspect is crucial. Will inter-individual variations impact the EEG-to-Language translation performance of the proposed method?

---

> ### Author Response · Authors · 2023-11-14
>
> ### Question 1:
> Quantization process of the EEG token is unclear. Whether the spatial arrangement of the EEG sensors played any roles in the process?
> ### Response:
> We use the public ZuCo dataset in our experiment. The ZuCo dataset segments the raw EEG wave using the eye fixation for each word and converts each segmented EEG wave into a total of 8 frequency bands.
>
> Originally, they used a 128-channel EEG cap to collect the EEG signal, but they removed 23 channels which resulted in a total of 105 EEG channels. As a result, each word-level EEG segment is processed into a 105*8=840 size frequency embedding. The maximum number of words in a sentence is set to 56. We therefore use the 56-word-level EEG embeddings as input tokens for the Conformer network.
>
> The Conformer network is a transformer-like architecture that outputs the same number of tokens as the input. Afterward, the output tokens from the Conformer, denoted by $h$ in our paper, will be quantized by a vector quantizer, and each token from the Conformer will be converted to a discrete token $b$. These discrete tokens will be used as input to the context transformer where the query prompt will interact with them via a cross-attention layer.
>
> Finally, the output of the context transformer (i.e., the mediate layer coding) can be sent to the LLM to get the final translation outputs.
>
> For the spatial arrangement of the EEG sensors. Since we are using a public dataset, the arrangement of the 105 EEG channel is fixed. Therefore, the effect of different sensor arrangements is not the research focus of this paper. However, we think it would be an interesting topic in the future works.
>
> ### Question 2:
> Question about the difference between BELT-1 and BELT-2.
> ### Response:
> Thank you for your feedback. We have briefly mentioned the different between BELT-1 and our BELT-2 model. We will provide more details in the final version. The limited discussion and comparison between the BELT-2 and BELT-1 models in our paper were primarily constrained by page limitations. The key design improvements in BELT-2 over BELT-1 can be summarized as : 1) **Enhanced Alignment**, 2)  **Muti-Task Flexibility**, and 3) **Integration with Pretrained LLMs**.
>
> ### Question 3:
> Questions about the 'EEG prompots' and the Mediate layer coding.
> ### Response:
> We used a BART model to initialize the context transformer. This makes the context transformer a bidirectional model that captures global information from the EEG tokens. However, participants read through a sentence according to the order they prefer and will also skip some words during the reading. As a result, we utilize the mediate layer coding as a 'prompt' to LLMs and use the finetuned prefix to instruct the LLMs to rearrange the disorder language information and generate more natural translation sentences.
>
> Last but not least, we will add illustrative information about the mediate layer coding and the EEG prompts to the figures to provide a clearer idea of the mediate layer coding.
>
> ### Question 4:
> Question about the dataset split and subject variations.
> ### Response:
> Thanks for the question. Currently, the experimental setup splits the dataset on a cross-training sample basis for a fair comparison with other methods. In response to the cross-subject splitting in the question and to explore whether inter-individual variations will impact the EEG-to-Language translation performance of our model, we additionally conducted the experiment in a cross-subject setting (shown in the table below). In this experiment, we leave one subject out as a test set and train the model on all other subjects. The performance is illustrated in the following table:
>
> | test subject | bleu-1 | bleu-2 | bleu-3 | bleu-4 | r    | p    | f1   |
> | ------------ | ------ | ------ | ------ | ------ | ---- | ---- | ---- |
> | ALL          | 43.1   | 25.6   | 15.2   | 9.2    | 30.6 | 34.3 | 32.2 |
> | ZPH          | 50.4   | 33.3   | 22.6   | 15.6   | 36.2 | 40.8 | 38.2 |
> | ZMG          | 50.5   | 33.2   | 22.3   | 15.4   | 35.1 | 39.2 | 36.9 |
> | ZKW          | 51.2   | 33.9   | 22.7   | 15.6   | 35.9 | 39.8 | 37.7 |
> | ZKH          | 51.4   | 34.5   | 23.2   | 16.0   | 36.8 | 40.8 | 38.6 |
> | ZKB          | 50.1   | 32.8   | 21.9   | 15.0   | 35.3 | 39.3 | 37.1 |
> | ZJS          | 48.4   | 30.8   | 19.7   | 12.9   | 33.6 | 37.7 | 35.4 |
> | ZGW          | 51.2   | 34.8   | 24.0   | 16.7   | 36.3 | 40.3 | 38.1 |
> | ZJN          | 48.0   | 30.5   | 19.7   | 13.0   | 33.6 | 37.8 | 35.4 |
> | ZDM          | 50.0   | 33.0   | 22.2   | 15.2   | 35.5 | 39.7 | 37.4 |
> | ZAB          | 50.1   | 33.0   | 22.3   | 15.3   | 34.6 | 38.6 | 36.4 |
>
> The table shows that the subject variant on the ZuCo dataset doesn’t have a significant impact on the EEG-to-Language translation performance of our model. We will include this ablation experiment in the supplementary in the final version.

---

> ### Author Response · Authors · 2023-11-21
>
> ### Extended response to Question 3:
> In the latest version of the paper, we have improved the readability and provided a clearer explanation of the mediate layer coding. For clarity, we rename the output of the Q-Conformer as the Mid-Layer Coding as these output embeddings represent a crucial **midpoint** between the EEG Encoder and LLM. Please refer to Section 2.3 and Figure in the current version of the paper for more details.
> ### Extended response to Question 4:
> In the latest version of the paper, we have included the cross-subject experiment in the main paper (Page 7 last paragraph, and Figure 6). Please refer to the current paper for more details.

---

> > ### Comment · Reviewer_CXNw · 2023-11-23
> >
> > Thanks to the authors for their comprehensive responses to my concerns and for conducting additional experiments. As a result, I have increased my rating.

---

> > > ### Author Response · Authors · 2023-11-23
> > >
> > > We sincerely appreciate your support and swift response. Your acknowledgment and appreciation of our work mean a tremendous amount to us.

---

### Official Review · Reviewer_KLkP · 2023-11-08

**Soundness:** 3 good
**Presentation:** 2 fair
**Contribution:** 2 fair
**Rating:** 6
**Confidence:** 4

**Summary:**

This paper presents BELT-2 which learns to perform many EEG-to-language tasks in a multitask setting. Specifically, EEG-to-text, summarization, and classification are learned simultaneously. The architecture consists of an encoder, which is pre-trained with a reconstruction loss. Then, during training time, all objectives for all tasks are optimized for simultaneously. The model can choose to tailor its representations per task, conditioned on a query vector that is also passed as input per task. The authors find that both pre-training and multi-task learning improve performance over the baseline.

**Strengths:**

- Significance: for the tasks considered, the results represent a substantial improvement over existing work
- Novelty: There is novelty in the application to the EEG domain
- It's unclear whether BPE-level contrastive learning is a novel method, or novel in the sense that this is the first time that it has been applied to the EEG domain. Could the authors clarify? If it's novel to the field, then this would be a plus for the paper

**Weaknesses:**

- The broader application to the field of machine learning is limited. The key takeaway seems to be the effectiveness of multi-task learning.
- It seems the main difference between BELT and BELT-2 is the addition of multi-task learning. If this is the case, then the technical advancement may be on the modest side. Although, I would not say this is a large weakness.

**Questions:**

- The last sentence on the first paragraph of page 1 says that previous methods have not achieved a "general bridging between brain signals and languages." Can you say more precisely what this means? Does "general bridging" refer to the multi-task setting? Is that the main difference between BELT and BELT-2?
- I have a question about the input to the conformer. Page 3 says that the input is created by "segmenting the raw EEG waveform signal into windows using eye-tracking information." Are the segments of uniform length? If so, then why is the eye-tracking information necessary? If not, then does some sort of truncation or down-sampling occur?
- I have a question about the multi-task query described on page 4. It says that "we could easily extend the Q-conformer to a new downstream task by initializing a new set of query tokens to extract information related to the new task, obviating the need for training an entirely new model." But is this really a saving in time? Wouldn't the same amount of time be needed to train the new set of query tokens, even if you start with the existing model weights?
- In section 2.4, it says that "During the EEG representation learning stage, the Q-Conformer extracts, task-specific information from the EEG input signals." But this doesn't seem to be the case? Is Figure 2 left meant to depict the representation learning stage? If so, why aren't there any task-specific terms in the objective function?

## small things
- Figure 2 -- the captions for (upper right) and (bottom right) are switched
- To get forward quotes, use `` in latex
- page 8 typo: "briding" --> "bridging"
- typo: the speculative augmentation ablation is Figure 5, not Table 5

---

> ### Author Response · Authors · 2023-11-14
>
> ### Question 1:
> Unclear whether BPE-level contrastive learning is a novel method, or novel in the EEG domain.
> ### Response:
> To the best of our knowledge, there is no prior work employing BPE-level contrastive learning in a manner similar to our approach.
>
> We conducted ablation experiments on the BPE in Table 1 in the main paper. We move the comparison for the BPE here for your convenience. Results in the table below show that the introduction of BPE-level contrastive learning brings significant improvement to the BLEU scores, reaching 43.06(+1.49), 25.57(+1.55), 15.16(+1.36), 9.17(+0.57). We will improve readability and refine the paper in the final version.
>
> | BPE Contrastive | BLEU-1 | BLEU-2 | BLEU-3 | BLEU-4 |
> | --------------- | ------ | ------ | ------ | ------ |
> | No              | 41.57  | 24.02  | 13.8   | 8.06   |
> | Yes             | 43.06  | 25.57  | 15.16  | 9.17   |
>
> Previous works on visual-language alignment are employing sentence-level contrastive learning [1,2]. However, due to the scarcity of EEG-language pairs, we need to incorporate stronger guidance from the language modality during training. Consequently, we consider that the use of BPE-level contrastive learning constitutes a novel method for aligning a modality with language when faced with a limited number of training sample pairs.
>
> [1] Image as a Foreign Language: BEiT Pretraining for Vision and Vision-Language Tasks.
>
> [2] Blip-2: Bootstrapping language-image pre-training with frozen image encoders and large language models.
>
>
> ### Question 2:
> Question about the "general bridging between brain signals and languages.". The main difference between BELT and BELT-2?
> ### Response:
> Here, "general bridging between brain signals and language" means this work is the first to achieve multi-task EEG decoding and is able to leverage the generative capacity of the existing LLMs.
>
> This includes being able to enhance the general understanding of brain signals by learning from different tasks while being able to decode multiple information from EEG signals (e.g., sentiment or summary). Our work signatures an important step to link the brain signal into existing multimodal understanding models which has the potential for boarder applications.
>
> The main difference between BELT-1 and BELT-2 is that BELT-1 is a single-task architecture while BELT-2 is a multi-task architecture. Our model accomplishes multi-task decoding using query prompts, a strategy aligned with current Language Models (LLMs). Also, BELT-2 allows us to directly combine various LLM (e.g., T5 and LLAMA2, PEGASUS) with our EEG Encoder to enhance the translation quality by merely tuning the prefix tokens. We are also the first work to provide a study (Table 5) of combining EEG encoders with different state-of-the-art LLMs.
>
> ### Question 3:
> Are raw EEG wave segments of uniform length? What is the use of eye-tracking information?
>
> ### Response:
> In our experimental setup, we leverage the ZuCo dataset [1]. Notably, this dataset incorporates eye-tracking data, recording eye fixations on individual words. The fixation duration for each word varies significantly, and some words may be omitted during reading, leading to EEG segments of different lengths.
>
> Although the length for each word-corresponding EEG signal is different, the ZuCo dataset performs a frequency transformation operation to extract a fixed number of frequency values from each EEG segment for each channel.
>
> [1] ZuCo, a simultaneous EEG and eye-tracking resource for natural sentence reading.
>
> ### Question 4:
> Question about "we could easily extend the Q-conformer to a new downstream task ... "
> ### Response:
> Here, we don’t need to retrain a new model or the query tokens. For the multi-task setting of our Q-conformer model, we trained 3 tasks together and implemented a sampling strategy to select a task for each training batch.
>
> ### Question 5:
> In section 2.4, it says that "During the EEG representation learning stage, the Q-Conformer extracts, task-specific information from the EEG input signals." But this doesn't seem to be the case? Is Figure 2 left meant to depict the representation learning stage? If so, why aren't there any task-specific terms in the objective function?
>
> ### Response:
> Figure 2 is meant to depict the feature extraction process of the Q-Conformer. In Figure 2 (left), our primary intention was to illustrate that the query prompts will query task-specific information from the EEG tokens via a cross-attention layer in the Context Transformer.
>
> As mentioned in the supplementary, the query prompt and the Q-Conformer are trained with the objective function mentioned in the appendix. We will make refinement to the figure in the final version.

---

> ### Author Response · Authors · 2023-11-21
>
> ### Extened Response to Question 5:
> In the latest revision, we have improved the readability of the paper. In particular, we revised the original Figure to give a more precise and clearer illustration of the Q-Conformer architecture. Please refer to the Figure 2 in the current paper. In Figure 2, we show the overall structure of the Q-Conformer. It consists of a discrete conformer, a context transformer (C-Former), and a query prompt. The input EEG embeddings (EEG embed) are first processed by the conformer into continuous EEG tokens. A vector quantizer is then used to discretize
> the EEG tokens. Then, a query prompt interacts with the discrete EEG token via the cross-attention layer from in the C-Former to extract task-specific context information from the discrete EEG tokens. While Figure 2 mainly depicts the interaction of submodules in the Q-Conformer, Figure 3 and Section 2.2 gives a more specific illustration of how the query prompt is trained. We summarize the first training stage as the EEG-to-language alignment learning stage where we use the EEG-to-language matching (ELM) objective as the 'base' task in this alignment learning stage. When training on ELM, the query prompt is trained to become task-specific. For multi-task training, as depicted in Figure 5, we assign a query prompt to each decoding task, so that each is trained by the objective of a task (e.g., translation, summary, or sentiment classification). Please refer to the current paper for more details and consider raising the rating for our work.

---

### Official Review · Reviewer_8fpS · 2023-11-08

**Soundness:** 3 good
**Presentation:** 3 good
**Contribution:** 2 fair
**Rating:** 5
**Confidence:** 4

**Summary:**

This paper introduces BELT-2, a multi-task model specifically designed to enhance both EEG signal encoding and decoding performance. BELT-2 incorporates byte pair encoding (BPE)-level EEG-language alignment and seamlessly integrates multi-task training and decoding within the EEG domain.

**Strengths:**

a)	It’s the first work of multi-task brain decoding by bridging the Q-Conformer EEG encoder and LLMs.
b)	It outperforms the baseline models, demonstrating superior performance.

**Weaknesses:**

a)	The interaction of all the components in Equation 2 is unclear, and it remains uncertain whether the introduction of certain hyperparameters is necessary.
b)	The organization of the logit structure in the paper appears somewhat disordered, exemplified by an error in the figure caption of Figure 2. I would recommend swapping the content in the upper right and bottom right sections of the figure. Furthermore, the explanation of loss functions on Page 3 does not align consistently with Equation 2.
c)	Lack of Clarity: The description of "Multi-task Query" in the paper is unclear. It is not clear how the query prompt is trained and whether a new task requires complete retraining. Furthermore, there is a lack of clarity regarding how the Frequency domain EEG embedding e is transformed into continuous EEG tokens h, i.e., the learning process of the conformer model E(.), and how it is subsequently transformed into word-level EEG representations.
d)	Lack of Specificity: If EEG representations are employed using a contrastive learning approach, the method proposed in Section 2.3 appears to be a conventional operation.
e)	Unclear Ablation Experiments: The paper does not provide a clear description of the ablation experiments, particularly the absence of ablation on BPE-level Contrastive learning and key components of Formula 2.

**Questions:**

See weaknesses

---

> ### Author Response · Authors · 2023-11-14
>
> ### Question 1:
> The interaction of all the components in Equation 2 and hpyerparameters is unclear.
> ### Response
> We combine the loss terms in Equation 2 using a weighted combination as mentioned in the supplementary (page 13 to 14). The combination of these loss terms is a common operation for training a Vector Quantised (VQ) encoder. Specifically, $L_{cm}$, and $L_{cb}$ are the default losses for training the discrete codebook as illustrated in VQVAE paper [1]. The $L_{div}$ is a term to encourage diversity in the usage of the discrete codebook entries which is also frequently used in training a VQ encoder, an example of using the diversity loss is in the wav2vec2.0 paper [2]. $L_{recon}$ is the necessary term used for training a reconstructive encoder as in the latent diffusion model [3]. We also provided ablation experiments on different hyper-parameter settings in the supplementary (Page 17).  We will organize these information from the supplementary to the main paper to improve the readability for boarder audiences.
>
> ### Question 2:
> Not clear how the query prompt is trained and whether a new task requires complete retraining.
> ### Response
> The Q-Conformer model does not require a complete retraining. We borrow the idea of query prompts from the recent visual-language model BLIP-2. The query prompt is able to select and extract information from the EEG tokens through a cross-attention layer in the context transformer.
>
> As mentioned in Page 4, the Q-Conformer efficiently adapts to new downstream tasks by initializing a set of query tokens, enabling the extraction of task-specific information. This adaptation is achieved through training a query prompt using the objective function tailored to the specific task. Given that BELT-2 is a multi-task model, individual tasks are trained concurrently without the need for complete model retraining.
>
> ### Question 3:
> How the Frequency domain EEG embedding is transformed into continuous EEG tokens.
> ### Response
> As mentioned in the paper (Page 3), EEG data from the ZuCo dataset is first segmented according to the gaze fixation on each word using eye-tracker information. A frequency domain transformation is applied to each EEG segment to extract value from 8 frequency bands. The ZuCo dataset contains 105 EEG channels. These frequency bands from the 105 channels form an equal-length word-level EEG embedding (size=840).
>
> Similar to previous works [4], the word-level EEG embedding is processed by a transformer-like EEG encoder that processes the word-level EEG embedding. We use the Conformer model to process the word-level EEG embeddings. After processing by the Conformer model, a vector quantizer to discretize the output EEG tokens into discrete tokens.
>
> ### Question 4:
> Lack of Specificity for Section 2.3.
> ### Response
> The reason why we adopt the negative contrastive loss is that the difference in human brain waves when looking at different words is very small, increasing the difficulties of EEG decoding. Therefore, we use the negative contrastive loss to enlarge the differences among EEG tokens and consequently enhance the final translation performance. The method of using negative contrastive loss among encoded tokens is also employed in the training of speech encoders [2].
>
> ### Question 5:
> Unclear Ablation Experiments: The paper does not provide a clear description of the ablation experiments, particularly the absence of ablation on BPE-level Contrastive learning and key components of Formula 2.
> ### Response
> We provided ablations about the BPE-level contrastive learning (Table 1), about using pretrained EEG encoder for downstream tasks (Table 3 and Table 4), about connecting the EEG encoder with different LLMs (Table 5), and about hyper parameters (Figure 9) in our main paper and in the supplementary. Due to the page limit, we moved the rest ablation experiments to the supplementary. We will reorganize more ablation information in the final version of the paper.
>
> Regarding to the effect of BPE-level contrastive learning, **we have already provided this ablation study in Table 1**.
>
> | BPE Contrastive | BLEU-1 | BLEU-2 | BLEU-3 | BLEU-4 |
> | --------------- | ------ | ------ | ------ | ------ |
> | No              | 41.57  | 24.02  | 13.8   | 8.06   |
> | Yes             | 43.06  | 25.57  | 15.16  | 9.17   |
>
> This ablation shows the performance of our model with or without the BPE-level Contrastive learning term. Results show that the introduction of BPE-level contrastive learning brings significant improvement to the BLEU scores, reaching 43.06(+1.49), 25.57(+1.55), 15.16(+1.36), 9.17(+0.57).
>
> [1] Neural discrete representation learning.
>
> [2] wav2vec 2.0: A framework for self-supervised learning of speech representations.
>
> [3] High-resolution image synthesis with latent diffusion models.
>
> [4] Open vocabulary electroencephalography-to-text decoding and zero-shot sentiment classification.

---

> > ### Author Response · Authors · 2023-11-21
> >
> > ### Extended Response to Question 1:
> > In the latest revision, we have improved the readability of the paper and provided a clearer introduction to the VQ loss term in Equation 2 in the main paper. Also, we include more details of the training process, loss function, and combination of different loss functions in Figure 3, Section 2.2, Section 2.3, Section 2.4, Appendix C.2 and C.3. Please refer to the current main paper for a more detailed introduction of the training and loss function used in this paper.
> >
> > ### Extended Response to Question 2:
> > In the latest revision, we improved the descriptions for training the query prompts in Section 2.2 and Section 2.4. In particular, we train a base query prompt by the EEG-to-language matching objective in the EEG-to-language alignment learning stage. In multi-task training, we train three tasks simultaneously. So, each task is instructed by a query prompt. Each query prompt will be trained by the objective function of the corresponding task. This is further illustrated in Figure 7 in the paper. A complete retraining is not required since we are a multi-task framework, we don't retrain a complete new model for each task but use the same model for multiple decoding tasks. Please refer to the current paper for more details.
> >
> > ### Extended Response to Question 4:
> > We have improved readability and understanding for a general audience in the latest revision. In particular, the negative contrastive learning loss is more clearly explained in Section 2.2 in the latest revision which is used to improve the distinction between discrete EEG tokens (page 4 last paragraph).

---

> > > ### Comment · Reviewer_8fpS · 2023-11-23
> > > **Increase rate to 5**
> > >
> > > Dear authors, thanks for your time and efforts in addressing my comments. Although some concerns remain, lots of points are clarified. I'll increase my rating to 5.

---

> > > > ### Author Response · Authors · 2023-11-23
> > > >
> > > > We are truly grateful for your time, expertise, and the confidence you've placed in our work. Your heightened rating empowers us to strive for greater excellence in our ongoing endeavors.

---

### Official Review · Reviewer_psny · 2023-11-14

**Soundness:** 3 good
**Presentation:** 3 good
**Contribution:** 3 good
**Rating:** 5
**Confidence:** 3

**Summary:**

This paper presents a novel multi-task model, called BELT-2, to enhance both encoding and decoding performance from EEG signals. The experimental results conducted have shown the effectiveness of the proposed method.

**Strengths:**

- The idea of the proposed method to bridge the Q-Conformer EEG encoder and LLMs seems interesting.
- The application to EEG data seems novel.
- The proposed method outperforms some state-of-the-art methods for various tasks.
- The paper is clear and well-structured.

**Weaknesses:**

- The discussion on the difference between BELT and BELT-2 is not sufficient.
- Some details related to the training and the loss function are not provided.

**Questions:**

It is suggested to highlight the difference between BELT and BELT-2, and to provide more precision on the training details and the loss function.

---

> ### Author Response · Authors · 2023-11-14
>
> ### Question 1:
> Difference between BELT and BELT-2.
> ### Response 1:
> Thanks for your question. The original paper mentioned the differences between these two models on page 2. Key design improvements in BELT-2 over BELT-1 can be listed as follows:
>
> 1. **Enhanced Alignment**: BELT-1 applies contrastive learning on word and sequence levels to bootstrap EEG encoder training. It has significant shortcomings: 1. Sequence-level alignment is compared coarser and would increase the difficulties for learning the exact relationship between EEG tokens and the corresponding word. 2. Word-level alignment would lead to a decreased performance in the decoding of rare words or out-of-training-vocabulary words when training data size or training vocabulary is limited.
>
>     BELT-2 introduces a significant improvement through BPE-level contrastive learning. It operates at a more granular level than word-level alignment as it breaks down words into smaller size of root words.  It is especially useful for handling rare words or words out of the vocabulary of the training set because of the shared subword units. Experimental results in Table 2 improvement lead to a clear improvement in the open-vocabulary setting.
>
>     This discussion is compressed due to the page limit. We will improve the writing to emphasize this point in the final version.
>
> 2. **Muti-Task Flexibility**: BELT-1 follows the same setting of all previous works that only explore brain-to-text translation, a single task.
> BELT-2 makes a significant improvement and extends the brain decoding to the multi-task setting for the first time.
> The multi-task setting makes the decoding supervision denser and also enhances the learning process.
> Additionally, it explores seamlessly switching between tasks by employing different query prompts, reducing computational resource requirements, and utilizing knowledge from diverse tasks.
>
> 3. **Integration with Pretrained LLMs**: BELT-2 offers the ability to connect the EEG encoder to a pre-trained Language Model (LLM), such as T5 or LLAMA2. This work provides a detailed comparison of different LLMs. It also includes an intuitive comparison between single-direction and bi-direction LLMs for follow-up researchers.
> This integration allows BELT-2 to leverage the strong generative capacity of pre-trained LLMs, providing enhanced performance.
>
> ### Question 2:
> More precision on the training details and the loss function.
> ### Response 2:
> Sorry for any inconvenience caused, due to the page limit, we moved some training details and the loss function to the appendix.
>
> For the training of the EEG encoder as illustrated in the appendix, i.e., the Q-Conformer, we used a weighted combination of the VQ loss, BPE-level contrastive loss, and the negative contrastive loss. For multi-task training, we trained all tasks together using a random sampling method on tasks. To bridge the trained Q-Conformer with the LLM decoder, we froze both the Q-Conformer and the LLM decoder and used the prefix-tuning method to train a set of prefix prompts to 'instruct' the LLM to decode fluent sentences from the output tokens of the Q-Conformer model. For the loss function used to train the translation task and the summarization task, we used a common machine translation loss, as in the BART paper [1]. For the sentiment classification task, we only used cross-entropy loss to train the last output token of the Q-Conformer, similar to the method in the XLNet paper [2].
>
> We will further concise the writing to include more information in the main paper.
>
> [1] Lewis, Mike, et al. "Bart: Denoising sequence-to-sequence pre-training for natural language generation, translation, and comprehension." arXiv preprint arXiv:1910.13461 (2019).
>
> [2] Yang, Zhilin, et al. "Xlnet: Generalized autoregressive pretraining for language understanding." Advances in neural information processing systems 32 (2019).

---

> > ### Comment · Reviewer_psny · 2023-11-20
> >
> > Thank you for addressing my comments.

---

> > > ### Author Response · Authors · 2023-11-21
> > >
> > > ### Extended Response to Question 1:
> > > Due to the page limit, we could only summarize the differences between BELT-2 and BELT-1 in the paper. In our latest revision of the paper, we have emphasized the difference between BELT-2 and BELT-1 in three key aspects. (1) BELT-1 is designed for a single task (e.g., the translation or the sentiment classification) while BELT-2 is for multi-task decoding (Page 2, line 8-9). (2) Compared to BELT-1, BELT-2 aligns EEG with language in a more fine-grained subword level (page 4). (3) Compared to BELT-1 and all other models, BELT-2 is the first framework that achieves the bridging between the EEG encoder and LLM (Page 2, second paragraph).
> > >
> > > ### Extended Response to Question 2:
> > > We have provided more precise training details and loss functions in Figure 3, Section 2.2, Section 2.3, and Section 2.4 in the latest revision of the paper. Please refer to the current main paper for more details. Also, additional details on training are also provided in the supplementary in Appendix C.2 and C.3. We also significantly improved the main paper for better readability. Please reconsider raising the rating for our paper as it signatures a significant step
> > > forward in integrating human brain signals with LLMs. This paper also presents solid experimental results with open-source code for ensuring reproducibility. We believe our research could have the potential to inspire further exploration and innovation in this research field.

---

> > > ### Author Response · Authors · 2023-11-23
> > >
> > > Dear reviewer,
> > >
> > > As your concerns and queries have been diligently addressed, would you consider raising the rating for our paper?  We have made immense efforts to enhance the quality and depth of our work. Your support in acknowledging these enhancements would be immensely appreciated.
> > >
> > > Best regards,
> > > Authors

---

### Meta-Review · Area_Chair_j7TV · 2023-12-19

**Metareview:**

Note that the authors broke confidentiality by stating "Dewave - Our pioneering endeavor to decode language straight from Raw EEG Waves. This was accepted by NIPS[sic] 2023." on their otherwise anonymous code repository. https://anonymous.4open.science/r/BELT-2-0048/README.md

This refers to "DeWave: Discrete EEG Waves Encoding for Brain Dynamics to Text Translation" by Yiqun Duan, Jinzhao Zhou, Zhen Wang, Yu-Kai Wang, Chin-Teng Lin. Recently presented at NeurIPS 2023 https://openreview.net/forum?id=WaLI8slhLw

Reviewers were by and large skeptical of the work. Critical details of the model are missing from the main work. Multiple reviewers had difficulty understanding the work and in particular the Q-Conformer. Reviewers were also skeptical about the technical advance of BELT-2 (this manuscript) relative to BELT-1 (the author's prior work, given that confidentiality was broken). The technical contribution of BELT-2 appears to be minimal compared to BELT-1. This is further complicated by the fact that the authors have withdrawn BELT-1 because of a technical flaw (teacher forcing at test time) which also affects the DeWave publication.

On a personal note. I want to urge the authors not to make comments to the AC of the form "Such comments are not only malicious, but they harm the atmosphere of academic discussion at ICLR." ACs look for reasons to accept. Focusing on positive attributes and the strength of your work rather than accusations of malicious behavior by reviewers is far more productive.

In summary, this manuscript cannot be accepted as is. It's relationship to prior work is unclear, its method is not described well enough to be easily understood.

**Justification For Why Not Higher Score:**

Unclear method, key missing details, a small technical advance over prior work, and a complex relationship to prior work.

**Justification For Why Not Lower Score:**

N/A

---

### Decision · Program_Chairs · 2024-01-16

Reject